# Identification and 3D modeling of bioactive peptides from *Lactobacillus brevis* RAMULAB49 protein hydrolysate with *in silico* ERK1 phosphorylation inhibition activity targeting diabetic nephropathy

Reshma Mary Martiz[1,2☯], Ramith Ramu[1*☯], Hemalatha Nambisan[3], Ameer Suhail[4], Mohammad Raish[5], Shashank M. Patil[1], Ashwini P.[2*], Mahesh B.[6], Maciej Przybyłek[7], Piotr Bełdowski[8], Alina Sionkowska[9], Kefeng Li[10], Xijun Tang[11*]

1 Department of Biotechnology and Bioinformatics, JSS Academy of Higher Education & Research, Mysore, Karnataka, India, 2 Department of Microbiology, JSS Academy of Higher Education & Research, Mysore, Karnataka, India, 3 St. Aloysius College of Management and Information Technology, Kotekar, Karnataka, India, 4 NeST Digital Private Limited, Kochi, Kerala, India, 5 Department of Pharmaceutics, College of Pharmacy, King Saud University, Riyadh, Saudi Arabia, 6 Department of Chemistry, JSS Academy of Technical Education (Affiliated to Visvesvaraya Technological University, Belagavi), Bengaluru, India, 7 Department of Physical Chemistry, Pharmacy Faculty, Collegium Medicum of Bydgoszcz, Nicolaus Copernicus University in Toruń, Bydgoszcz, Poland, 8 Institute of Computer Science, Kazimierz Wielki University in Bydgoszcz, Bydgoszcz, Poland, 9 Department of Biomaterials and Cosmetic Chemistry, Faculty of Chemistry, Nicolaus Copernicus University in Toruń, Toruń, Poland, 10 Faculty of Applied Sciences, Macao Polytechnic University, Macau, China, 11 Zhuhai Hospital Affiliated to Faculty of Chinese Medicine, Macau University of Science and Technology, China

☯ These authors contributed equally to this work.
* ramith.gowda@gmail.com (RR), tangxijun123@126.com (XT), ashwinip@jssuni.edu.in (AP)

## Abstract

Diabetic nephropathy (DN) poses a significant health challenge, necessitating novel therapeutic approaches. In this study, we isolated proteins from cell-free supernatant (CFS) from the culture of the lactic acid bacteria *Lactobacillus brevis* RAMULAB49 strain. The proteins were subjected to simulated *in vitro* gastrointestinal digestion using gut enzymes – pepsin, pancreatin, and trypsin. The hydrolysates were filtered using 3kDa threshold ultra-centrifugal filters and were desalted using C18 disks. This was followed by nLC-ESI MS/MS tandem mass spectrometry-based identification of peptides, leading in the identification of a of 258 unique peptides across three enzyme combinations. The resultant sequences were made into peptide library construction based on their, bioactivity scores, allergenicity, toxicity, and antidiabetic potential, a total of 10 peptides was constructed and modeled in 3D. On the other hand, 266 DN associated genes were identified using a network pharmacology approach. The resultant protein-protein (PPI) network was analysed using the gene ontology (GO) and Kyoto Encyclopedia of Genes and Genomes (KEGG) pathway enrichment approaches, resulting in identification of critical pathways, ERK1, PI3K-Akt, EGRF and TNF signaling as significantly involved in DN, where, ERK1 emerging

**Data availability statement:** All relevant data are within the manuscript and its Supporting Information files.

**Funding:** The author(s) received no specific funding for this work.

**Competing interests:** The authors have declared that no competing interests exist.

as a key node due to its involvement in cell proliferation, inflammation, and fibrosis associated with DN. Top two 3D-modelled bioactive peptides were selected for interaction study with the target protein ERK1. Peptide TNEDPYTIDVES showed a strong binding energy of −9.9 kcal/mol, at the ATP-binding site and dynamics simulations confirmed the structural stability of this complex over 100 ns, showing consistent hydrogen bond interactions and RMSD values below 2.5 Å. These findings suggest that TNEDPYTIDVES may act as a competitive ERK1 inhibitor by occupying the adenine-mimicking ATP-binding cleft, thereby interfering with phosphorylation activity. This integrative approach highlights *L. brevis* RAMULAB49 strain derived peptides as promising candidates for the development of peptide-based therapeutics target and could pave the way for new drug development treating diabetic nephropathy.

## 1. Introduction

Diabetic nephropathy (DN), generally referred to as diabetic kidney disease (DKD), is a kidney disease that is a complication of type 1 and type 2 diabetes. It occurs when poorly controlled diabetes damages the blood vessels in the kidneys that filter waste from the blood. This entails the disruption of the complex network of renal microvasculature and can lead to kidney damage [1–3]. About 30% of people with diabetes are said to have diabetic nephropathy (DN), according to statistics. For public health systems, the high frequency of DN among diabetes patients is a considerable burden [4,5].

The pathogenesis of diabetic kidney disease is multifactorial including insulin resistance, hyperglycaemia, hemodynamic, physiological, autoimmune, and inflammatory processes [5]. The most common morphological abnormalities in the early stages of DN are glomerular hypertrophy, podocyte depletion, tubular damage, glomerular basement membrane (GBM) thickening and mesangial matrix expansion [6]. DN has a complicated and poorly understood pathophysiology, which results in inadequate therapy outcomes. It has been established that standard therapy with rigorous blood pressure and blood glucose management is insufficient to stop DN from progressing to end-stage renal disease (ESRD) and mortality [4,7].

Recent studies suggested that abnormal progression of the renal tissue has been the core issue related to DN. The renal tissue dysfunction has been attributed to the ERK1 or ERK11 signalling pathway, which is linked with renal morphogenesis and differentiation [8]. The activation of ERK1/2 has also been reported in the development of both nephron progenitors and renal tissue. However, the over-expression of ERK1/2 is reported with dysregulation of epithelial-mesenchymal transformation, resulting in renal fibrosis [9]. The reduced expression of ERK1/2 has been reported with reduced renal fibrosis and abnormal tissue progression. To date, only a few studies have targeted ERK1/2 expression for anti-diabetic therapies [10,11]. These drugs including angiotensin-converting-enzyme (ACE) inhibitors, angiotensin II receptor blockers (ARBs), and statins, have proven to be quite helpful in decreasing the early-stage development of DN.

Despite the availability of pharmacological options such as ACE inhibitors, ARBs, statins, and SGLT2 inhibitors, current treatment strategies primarily focus on controlling glycemic levels and blood pressure, rather than reversing or halting disease progression. These agents are often accompanied by undesirable side effects including hypotension, hyperkalemia, allergic reactions, and drug interactions, and they fail to adequately address the complex molecular pathways involved in DN pathogenesis [12,13]. As such there is a need to novel therapy alternatives with enhanced targetability, bioavailability, effectiveness, and capable of modulating specific signaling mechanisms such as ERK1-mediated fibrosis and inflammation. Thus, the demand for natural alternatives to food and dietary supplements is rising [14,15]

Since their positive impact on a variety of physiological and biochemical processes, the study of peptides has made tremendous progress over the past few years. These peptides have piqued researchers' interest due to their potential as medicines or functional ingredients for various diseases. They've shown promise as enzyme regulators, receptor modulators in oncology and inflammation, and essential components in antibiotic-infused medications [14].

Bioactive peptides, especially those derived from natural sources has accelerated in recent years, with numerous studies are exploring their potential in treating various diseases [16]. Bioactive peptides derived from marine resources, such as antioxidant peptides from Miiuy croaker swim bladders and Hizikia fusiformis, have been reported to exhibit potent bioactivity [17–19]. Some of these peptides have shown potential in improving gut health and modulating immune responses which are important in management of DN. Despite these study on bioactive peptides from different sources, the potential of LAB derived peptides remains relatively underexplored for treating conditions like DN.

Lactic acid bacteria, offer promising advantages. These include high target specificity, low toxicity, biodegradability, and potential for modulating multiple disease-related pathways. Their natural origin also aligns with the growing interest in food-based therapeutics and functional nutraceuticals [20]. Since LAB from various sources around the world have been associated with profound antidiabetic activity, and peptides from LAB are yet to be explored for their pharmaceutical values against health ailments, LAB remains one of the prominent sources of bioactive peptides [21,22].

Various studies as indicated that LAB, may have beneficial effects on DN [23–25]. These microbes have been shown to improve gut health, reduce inflammation, and modulate immune response, which are important factors in the management of DN. Notably, *Levilactobacillus brevis* has exhibited anti-inflammatory properties, which could be crucial in managing conditions like diabetic nephropathy, where inflammation plays a significant role in kidney damage [23].

Building upon the authors' previous work [21], this study investigates the identification of bioactive peptides from the protein hydrolysates of *Levilactobacillus brevis* RAMULAB49 isolated from fermented pineapple (*Ananas comosus*), with a specific focus on their therapeutic potential in managing DN. While the previous work [21] has confirmed the antidiabetic efficacy of the *L. brevis* RAMULAB49 strain through the inhibition of carbohydrate-digesting enzymes like α-glucosidase and α-amylase, its role in modulating DN-specific molecular targets has remained unexplored. In this study, we first employed a network pharmacology-based approach to identify ERK1 as a key target involved in the pathogenesis of DN. Subsequently, peptides derived from *L. brevis* RAMULAB49 were profiled, and the most potent ERK1-targeting peptide was identified through molecular docking and dynamics simulations. This study is an *in silico* approach which paves the way for the advanced *in vitro* and *in vivo* validation studies to confirm the therapeutic potential of peptides. These findings suggest that *Levilactobacillus brevis*–derived peptides could offer a novel, targeted strategy against DN, although further investigation is needed to elucidate their precise mechanisms of action and establish appropriate dosage and delivery methods.

## 2. Materials and methods

### 2.1. Materials and reagents

The experimental part of the study included *Levilactobacillus brevis* RAMULAB49 strain culture, ammonium sulfate (HiMedia Laboratories Pvt. Ltd), a nanodrop spectrophotometer, pepsin, trypsin and pancreatin for *in vitro* gastrointestinal digestion (HiMedia Laboratories Pvt. Ltd), LC-MS/MS equipment (SAIF IIT Bombay), Thermo Proteome Discoverer 2.2 and BioPharma Finder Software 2.0 softwares maintained by Thermo Fisher Scientific Inc.

## 2.2. Preparation of *L. brevis* RAMULAB49 protein hydrolysates

The preparation of *L. brevis* RAMULAB49 was performed following the previously established protocol by the authors [21], which demonstrated the efficient release of bioactive compounds, including peptides. Given the proven efficacy of this method, the same protocol was adopted without modifications. The same bacterial strain (RAMULAB49) was used in this study, and all experimental conditions including growth parameters, protein extraction, *in vitro* digestion procedures were maintained as described earlier to ensure consistency and reproducibility.

Briefly, lactic acid bacteria *L. brevis* RAMULAB49A was grown in 1 litre of MRS broth at 37°C for 48–72 hours. The culture was centrifuged at 15,000 × g for 20 minutes at 4°C to pellet the cells and debris. The resulting cell-free supernatant (CFS) was subsequently passed through a 0.22 μm syringe filter to remove remaining live cells [21]. Further, CFS was diluted 1:1 with 3.5% KCl-HCl buffer (100 mM, pH 2.0) and subjected to in vitro digestion by adding pepsin (4% w/v), followed by incubation at 37°C for 4 hours. The enzymatic reaction was terminated by heating the mixture in a boiling water bath for 10 minutes, after which the pH was adjusted to 7.0 using 2 N NaOH. The digested contents were centrifuged at 10,000 × g for 45 minutes at 4°C to collect the hydrolysate as supernatant. The hydrolysate was further subjected to a second and third *in vitro* digestions under the same conditions, replacing pepsin with pancreatin and trypsin (4% w/v), respectively. The final hydrolysates were stored at −20°C until further analysis [26].

For peptide fractionation, the hydrolysates were filtered using Amicon® Ultra centrifugal filter units with a 3 kDa molecular weight cut-off (MWCO) by centrifugation at 8,000 × g for 30 minutes at 4°C to obtain permeates containing peptides smaller than 3 kDa. The permeates were then desalted using C18 solid-phase extraction (SPE) disks, which were preconditioned by passing 5 μL of 90% methanol in sterile water to activate the column material and remove impurities. Residual organic solvent was removed by washing the column with 5 μL of 3.5% KCl-HCl buffer (100 mM, pH 2.0), followed by equilibration with an additional 5 μL of the same buffer. The hydrolysate was then loaded, washed once with the buffer, and eluted using 4 μL of 50% acetonitrile in sterile water mixed with 1 μL of 3.5% KCl-HCl buffer. The permeates were collected in sterile vials were subjected to lyophilization, followed by nano-Liquid Chromatography-Electrospray Ionization-Tandem Mass Spectrometry (n-LC-ESI-MS/MS) analysis [26].

## 2.3. Identification of peptides using nLC-ESI MS/MS tandem mass spectrometry

Peptides and their corresponding proteins from the hydrolysates were identified using a Thermo Fisher Scientific Q-Exactive Plus Biopharma High-Resolution Orbitrap mass spectrometer coupled to an Easy n-LC 1200 nano-LC system. A total of 12 μL of each sample was loaded onto an Acclaim PepMap RSLC C18 analytical column (2 μm particle size, 100 Å pore size, 50 cm length) with an Acclaim PepMap 100 pre-column (100 μm × 2 cm, nanoViper) used for pre-concentration and desalting [26]. Mobile phase A consisted of 0.1% formic acid in Milli-Q water, and mobile phase B consisted of 80:20 acetonitrile:Milli-Q water with 0.1% formic acid. Peptides were separated with a linear gradient from 2% to 80% solvent B over 60 minutes at a flow rate of 300 nL/min. The nano-LC system was directly interfaced with the mass spectrometer, and MS/MS data were acquired using collision-induced dissociation (CID) with nitrogen as the collision gas, while singly charged ions were excluded from MS/MS analysis to prioritize multiply charged species. The raw data were processed using Thermo Proteome Discoverer 2.2 (Thermofisher Scientific, Massachusetts, USA) software and searched against the UniProt database corresponding to the sample organism (*Lactobacillus* spp.), with digestion parameters set to pepsin and pancreatin specificity. No fixed or variable modifications were included in the database search, as the goal was to map naturally occurring peptides from enzymatic hydrolysis without chemical alterations. The sequences with high confidence (>95%) and q < 0.01 were considered as potential hits. The intact mass identification and peptide mapping were done using BioPharma Finder Software 2.0 (Thermofisher Scientific, Massachusetts, USA). A fixed threshold strategy was applied (< 5.0 ppm with >95% confidence) for MS/MS auto-validation to ensure high-confidence peptide identification [26].

## 2.4. *In silico* physicochemical characterization, 3D modeling, and peptide drug library construction

The proteins and peptides identified were manually analyzed to generate separate lists of peptides and their corresponding proteins. Peptides exhibiting potential bioactivity were manually selected for further analysis. In the pre-processing step, peptides were segregated from their parent proteins. Manual screening of the peptide pool was carried out in multiple stages to identify candidates with pharmacological bioactivity, minimal toxicity, and low allergenicity. Initially, peptide sequences containing 5–20 amino acid residues and a molecular weight range of 0.4–2 kDa were preferentially selected, as peptides within this range are typically associated with higher bioactivity and greater resistance to enzymatic and physicochemical degradation [26].

Further, the physicochemical properties and stability of the selected peptides were assessed using the ProtParam tool available at https://web.expasy.org/protparam/ (Accessed on August 20, 2023). The allergenicity prediction of the peptides was performed using the bioinformatics tool AllerTOP, accessible at https://www.ddg-pharmfac.net/AllerTOP/index.html (Accessed on August 20, 2023), as described by [26]. ToxinPred (https://webs.iiitd.edu.in/raghava/toxinpred/index.html) (Accessed on August 20, 2023) and DBAASP (https://dbaasp.org/home) (Accessed on August 20, 2023) were utilized to predict the toxicity and antimicrobial properties of the peptides, respectively. The BIOPEP-UWM database (http://www.uwm.edu.pl/biochemia/index.php/en/biopep) (Accessed on August 22, 2023) developed by was employed to screen and evaluate the antidiabetic potential of the peptides. These computational tools and databases provided valuable insights into the characteristics and potential applications of the peptides in relation to allergenicity, toxicity, antimicrobial activity, and antidiabetic properties.

The peptide structures obeying the above-mentioned criteria were separated from the original peptide pool and were modelled to obtain the 3D structure using the i-TASSER server (https://zhanggroup.org/I-TASSER/) (Accessed on August 25, 2023) and refined using the ModRefiner web tool (https://zhanggroup.org/ModRefiner/) (Accessed on August 25, 2023). The accuracy of the predicted peptide models was re-assessed using the Ramachandran Plot analysis through the PROCHECK tool (https://www.ebi.ac.uk/thornton-srv/software/PROCHECK/) (Accessed on August 26, 2023) [27,28]. This validation process ensures the accuracy and high quality of the computations. The peptides selected from all the above-mentioned analyses were categorized as bioactive peptides and were used to create a functional dataset. This dataset can serve as a valuable resource for drug discovery targeting diabetes and its associated diseases or disorders. No data from this has been made publicly available.

i-TASSER server is an online platform used for protein structure and function predictions. It provides precise predictions of the molecular structure and biological activity of proteins derived from their amino acid sequences. It uses a locally installed meta-threading approach or LOMETS [29] to identify template proteins from the PDB database that display comparable folds or super-secondary structures. Further, it reassembles the fragments using Monte Carlo simulations and simulates the ab initio 3D model generation using PDB entries as the template. The model reassembles and verifies the structures beginning with the SPICKER cluster centroids [30]. In order to generate the full atomic model, REMO algorithm (reconstruct atomic model from reduced representation) was applied [31]. The function predictions are based on the consensus of top structural matches, which consider confidence score, structural similarity, and sequence identity [32]. Similarly, PROCHECK is an useful online tool that enables the checking of the quality of stereochemical features of protein structures. It generates PostScript plots that comprehensively evaluate geometry of a protein, including residue-by-residue detail. The tool is particularly useful for supporting NMR structure determination and validating the final results [28].

## 2.5. Human PPI network construction and DN therapeutic target identification

A total of six databases were used to construct three high-throughput human PPI networks and a DN-specific PPI network to gain insights into the pathophysiology of diabetic nephropathy. The human PPI networks were constructed using the entire dataset from HuRI (http://www.interactome-atlas.org/) (Accessed on August 28, 2023), HINT (http://hint.yulab.org/) (Accessed on August 28, 2023) and BioGRID (https://thebiogrid.org/) (Accessed on August 28, 2023). A total

of 52,548 binary protein-protein interactions were mapped by the Human Reference Interactome (HuRI) (Accessed on August 30, 2023). As a high-quality curated interactome resource of 8 databases, HINT mapped 119,526 interactions (accessed 23 March 2023). Similarly, the Biological General Repository for Interaction Datasets (BioGrid) contains 1,048,576 experimentally validated interactions (Accessed on August 30, 2023). Each protein in these databases was mapped to the relevant UniProt ID to increase robustness and eliminate network bias. Any protein and its associated interactions that could not be linked to a UniProt ID were disregarded from the study in order to ensure the accuracy of the data. In an effort to create a core human PPI network with a high level of confidence, the three distinct protein-protein interaction (PPI) networks were integrated and the overlapping interactions were kept. Using this method, a thorough and reliable network that can be used to research protein interactions and their functional consequences in human biology was created. OMIM (https://www.omim.org/) (Accessed on September 1, 2023), DisGeNET (https://www.genecards.org/) (Accessed on September 1, 2023) and GeneCards (https://www.genecards.org/) (Accessed on September 1, 2023) are three valuable resources for discovering genetic factors associated with the development of diseases. Genes associated with diabetic nephropathy were retrieved from the database with the keyword 'diabetic nephropathy'. Cytoscape was used to construct a large undirected DN-specific PPI network using STRING. The unconnected nodes were removed and connected nodes were retained in this study. Finally, we integrated the DN-specific PPI network into the primary human PPI network to inspect the intersections and perform a comprehensive study of disease-related modules and pathways.

Topological analysis of PPI networks reveals key proteins, modules, and pathways that are essential for comprehending cellular processes, disease mechanisms, and potential therapeutic targets. Functional modules in the PPI network were identified using the MCODE (Molecular Complex Detection) graph clustering algorithm [33]. The most compact cluster with the highest MCODE score was subjected to further analysis. For a comprehensive cluster identification, default cut-offs (0.2) were specified for the fluff and haircut parameters. Hub proteins were identified on the cluster. CytoHubba [34] and several topological algorithms were employed to generate five sub-networks, which were then integrated to form networks of significant proteins. These algorithms were Degree, Edge Percolated Component (EPC), Maximum Neighborhood Component (MNC), and Shortest Path Centralities, such as Closeness and EcCentricity. Highly significant genes were filtered out from the subnetwork based on the primary score file calculated by CytoNca [35].

## 2.6. KEGG, GO and DO enrichment analysis

Enrichment analysis uncovers highly functional terms or disease associations of the PPI network. Multiple enriched annotations can provide insights into the complexity and diversity of the biological system under study. This is because the gene set encompasses genes that participate in various biological pathways, regulatory networks, or functional modules. The findings provide essential clues for further examination and research into the underlying biological mechanisms. Gene ontology (GO), disease ontology (DO), and Kyoto Encyclopedia of Genes and Genomes (KEGG) pathway enrichment analysis was performed on the CytoNCA-generated critical sub-network. The ShinyGO 0.77 tool was used *to* perform each analysis while accounting for an FDR cut-off of 0.05.

## 2.7. Molecular docking

In this study, the structure of human ERK1 in complex (PDB ID: 4QTB) (380 residues) [36] was obtained from the RCSB PDB database(https://www.rcsb.org/) (Accessed on September 3, 2023). The Autodock 4.2 software was used for protein structure preparation and refinement. Heteroatoms and nonessential moieties were removed from the structure, followed by energy minimization. This process ensures that the protein structure is energetically favourable and suitable for accurate docking simulations. For the docking simulations, HADDOCK v2.4 (High Ambiguity Driven Protein-Protein Docking) was utilized as the docking method. HADDOCK is an information-driven docking tool that incorporates various information on binding sites and which performs semi-flexible docking by allowing flexibility in the side chains and backbone of the

atoms at the interface during the refinement stage. Thus, allowing the system to account for conformational adjustments upon peptide binding. [37].

The predefined parameters for the docking trials included a random exclusion of 50% of the "active interaction restraints" (AIRs). The standard setting of 2.0 for the number of random exclusion partitions, which represents the number of partitions used for random exclusion, was chosen. The pairwise backbone root mean square deviation (RMSD), with an adequate threshold of 5.0, was then used as the basis for the clustering study. Docking results were evaluated using various metrics, which includes, HADDOCK score, electrostatic energy, van der Waals energy, de-solvation energy, and buried surface area. Clusters were ranked accordingly. The Z-score of the best cluster was assessed for the reliability of the docking solution.

3D interaction analysis was constructed using Discovery Studio to enhance the reliability of the docking results. These diagrams shed light on the interactions and bonds that exist between protein-peptide complexes. By applying these techniques, the study intends to provide an in-depth knowledge of how protein and peptide molecules interact with one another as well as useful insights on their binding properties and their functional implications.

## 2.8. Molecular dynamics simulation

Simulations were performed with the CHARMM36 force field in GROMACS 2018, utilizing CHARMM-GUI for the considered in this study protein-peptide complexes (https://www.charmm-gui.org/) (Accessed on September 5, 2023) [38]. The protein structure was prepared for molecular dynamics simulations using the GROMACS program, employing the "pdb2gmx" script. To achieve efficient solvation of the molecular assemblies, simulations were conducted in a periodic dodecahedral unit cell, using the single point charge (SPC) model for water molecules. Next, to achieve a zero net charge in the system, Na+ ions were introduced into the solvated environment using the "gmx_genion" tool. All molecular assemblies underwent minimization employing the steepest descent algorithm, utilizing an energy step size of 0.01 over 50,000 iterations.

The simulation began with the NVT ensemble, maintaining a constant number of particles, volume, and temperature, for 1000 ps with a time step of 2.0 fs. This was followed by a 1000 ps equilibration using the NPT ensemble, which kept the number of particles, pressure, and temperature constant. Subsequently, each protein-peptide complex underwent 100 ns of all-atom molecular dynamics simulation. After equilibration, the solvated protein-peptide complexes were used to obtain the trajectories for radius of gyration (Rg), root mean square deviation (RMSD), root mean square fluctuation (RMSF), solvent accessible surface area (SASA), and hydrogen bonds (H-bonds) using GROMACS utilities.

## 3. Results and discussion

### 3.1. *In vitro* gastrointestinal digestion and Identification of peptides by nano-LC-MS/MS

To imitate the real digestion that dietary proteins go through in the gastrointestinal tract, the protein was digested using *in vitro* gastrointestinal simulation approach. This was done in order to find peptides that could potentially be bioactive that were concealed within the sequences of precursor proteins. The investigation sought to find peptides that might possibly have bioactive qualities by putting the protein samples to enzymatic digestion, which is similar to the natural digestive processes. The quantities of peptides in three separate samples—enzyme-digested P (pepsin), PP (pepsin and pancreatin), and PPT (pepsin, pancreatin, and trypsin)—were measured during the experiment (Supporting information S2 File). According to the measurements, the concentrations of the protein hydrolysates were 0.15 ± 0.07 mg/mL, 0.88 ± 0.02 mg/mL, and 0.75 ± 0.05 mg/mL, respectively (Supporting information S2 File). This quantification revealed information on the number of peptides produced during the digesting procedures **(Supporting information S1 File and S2 File)**.

Following digestion, the sequences and molecular weights of the peptides were determined using nano-LC-MS/MS (nanoscale liquid chromatography-tandem mass spectrometry). After acquiring the data from LC/MS analysis using Thermo Proteome Discoverer 2.2 software, the results were confirmed by comparing the data using UniProt database

(Supporting information S2 File). The protein database for lactic acid bacteria prepared after successful identification of query proteins with UniProt Accession Number.

The results showed that the pepsin-digested sample contained 61 different peptide sequences, with molecular weights ranging from 794.95 Da (Daltons) for the peptide LLAPPER to 3959.5 Da for the peptide TMMFVPGNNAGMVKDAGI-YGADSIMFDLEDSVSMSEK. Similar to this, 65 peptides with molecular weights ranging from 732.79 Da (KATGDNK) to 1664.90 Da (GEHNIVMTYATPGFK) were found in the pepsin and pancreatin-digested sample. Notably, 132 peptides with molecular weights ranging from 774.86 Da (NNISISK) to 3108.32 Da (ADFPNTFILGEAASANVNLAVDYTSQHNK) were found in the protein sample digested with a mixture of pepsin, pancreatin, and trypsin. These results illustrate the wide range of peptides produced by the simulated digestion process (Fig 1). The determined molecular weights and sequences provide important information about any potential bioactive peptides that may have been present in the initial protein sequences. Understanding the function of these peptides in various physiological processes and their prospective uses in the fields of nutrition and health are aided by this knowledge. The thorough investigation provides insight into the complex molecular alterations that take place during protein digestion and provides a framework for further investigation of the functions of bioactive peptides.

### 3.2. In silico peptide profiling

The selection of peptides involved meticulous consideration and involved a thorough computational analysis that taken account of a number of significant parameters. These selected peptides were required to satisfy some criteria, such as having an amino acid sequence with 8–20 residues and a molecular weight (MW) below 1500 Da. Smaller MW, especially below 5 kDa, has been associated with stronger inhibitory effects, as shown in research by **(Walther B and Sieber R. (2011) and Nong NT and Hsu JL (2022))** [39,40], among others. Additionally, findings from a study by **Jakubczyk et al., 2020** [41] highlighted the advantages of low molecular weight peptides, which support enhanced antioxidant and antibacterial properties. Furthermore, as evidenced by **Mirzaei et al., 2020** [35] study, peptides with an instability score below 40 were chosen because to their known stability. The GI stability of peptides is one of the most discussed concerns in peptide therapeutics [42]. Therefore, the development of GI-stable peptides is a considerable investment in peptide-based drug development. Since the instability score is below 40, these bioactive peptides are considered to be stable at the gastrointestinal environment [43]. These remarkable outcomes have been achieved without altering or the chain modification of

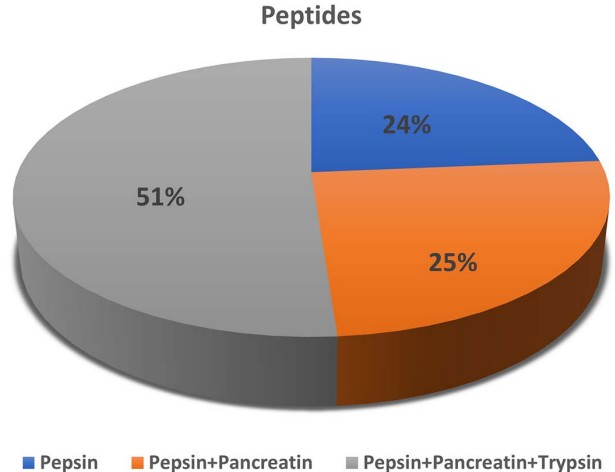

**Peptides**

■ Pepsin   ■ Pepsin+Pancreatin   ■ Pepsin+Pancreatin+Trypsin

**Fig 1. Distribution of peptides (in %age) in different *in vitro* gastrointestinal digestive modules.**

peptides, though there are several approaches available to improve peptide structure stability [44]. The selected peptides underwent additional screening after passing through the physicochemical property filter (**Supporting information** S1 File). Assessments for allergenicity, toxicity, and antibacterial activity were adequately satisfied by 10 peptides. Further analysis looked at possible antidiabetic qualities, and it was shown that α-glucosidase (AG) and Angiotensin Converting Enzyme (ACE) inhibitor activities were most frequently repeated. The 10 peptides that exhibited the best qualities were finally selected as the top candidates for further investigation after the rigorous selection process. In peptide therapeutics, allergenicity, toxicity, and bioactivity are the 3 main features that are being majorly considered [45]. The selected peptides are predicted to possess dual therapeutic roles including antimicrobial and antidiabetic activity. Peptides with multiple therapeutic activities with special reference to antimicrobial activity are essential since they need to withstand gut micro-biota and their bioactives [46]. The selected bioactive peptides also have a dual role in exerting antidiabetic activity – they have been predicted to inhibit 2 important enzymes including AG and ACE. AG is primarily linked with hyperglycemia and ACE constricts the blood vessels to increase blood pressure. Since both diabetes and blood pressure are linked and can be seen in obese people, the peptides could be used as suitable for obesity-linked diabetes mellitus too [47]. The peptides that stood out as potential contributors to desired results in future studies are shown in Table 1 as part of this selection. The heatmap for the selection of the peptides using all the defined screening criteria has been given in Fig 2.

### 3.3. PPI network construction and Functional analysis

The construction of protein-protein interaction (PPI) networks plays a crucial role in understanding the functional relation-ships between proteins and studying their cellular processes. While previous researches provided a significant starting point, PPI analysis offers more comprehensive and unbiased identification of potential targets by investigating the inter-actions between proteins within disease context [48,49]. Through this approach a various novel target that may not have been previously considered may be uncovered [50]. In a study by **(Safari-Alighiarloo et al., 2014 & Martiz et al., 2022)** [35,48], a comprehensive PPI network was constructed to explore the complex interactions and communication between proteins, providing valuable insights into biological pathways, molecular functions, and disease mechanisms. PPI analysis also validates these findings, reinforcing the rationale for their consideration as therapeutic targets [49–51].

The primary human PPI network in this study consisted of 6,866 nodes and 2,260 interactions, with an average of 2.731 neighbours per node. To investigate the genetic associations with DN, a search was conducted through databases such as OMIM, DisGeNET, and GeneCards, identifying a total of 4,005 genes linked to DN. Subsequently, a DN-specific

**Table 1. Peptide profiling of 10 selected peptides.**

| Sequence | MW [Da] | No of AA | Toxicity prediction | Allergenicity prediction | Structure stability | Antimicrobial Activity | Biological Activity |
|---|---|---|---|---|---|---|---|
| LSNRAAFFR | 1081.24 | 9 | Non-toxic | Non-Allergen | 8.89 | AMP | ACE inhibitor |
| VTDLDLTAEVVK | 1302.49 | 10 | Non-toxic | Non-Allergen | 5.62 | AMP | ACE inhibitor |
| FENHAVEVDELSR | 1544.64 | 13 | Non-toxic | Non-Allergen | 2.88 | AMP | AG inhibitor/ACE inhibitor |
| LGADATVPFDMTTK | 1466.67 | 14 | Non-toxic | Non-Allergen | 6.76 | AMP | AG inhibitor/ACE inhibitor |
| KSSLVTGQQLTGANK | 1531.73 | 15 | Non-toxic | Non-Allergen | 38.89 | AMP | ACE inhibitor |
| VTQGSINFAKSVAENYK | 1856.07 | 17 | Non-toxic | Non-Allergen | 24.62 | AMP | ACE inhibitor |
| AGTSFTIGSFNGDGWNSIK | 1959.1 | 19 | Non-toxic | Non-Allergen | 14.43 | AMP | ACE inhibitor |
| KMDLAELK | 947.16 | 8 | Non-toxic | Non-Allergen | −8.86 | AMP | ACE inhibitor |
| DLLIDAQDVQK | 1257.41 | 11 | Non-toxic | Non-Allergen | −15.52 | AMP | ACE inhibitor |
| TNEDPYTIDVES | 1382.4 | 12 | Non-toxic | Non-Allergen | −17.24 | AMP | AG inhibitor/ACE inhibitor |

**Note**: AMP: Anti-microbial Peptide, AG: α-glucosidase, ACE: Angiotensin Converting Enzyme.

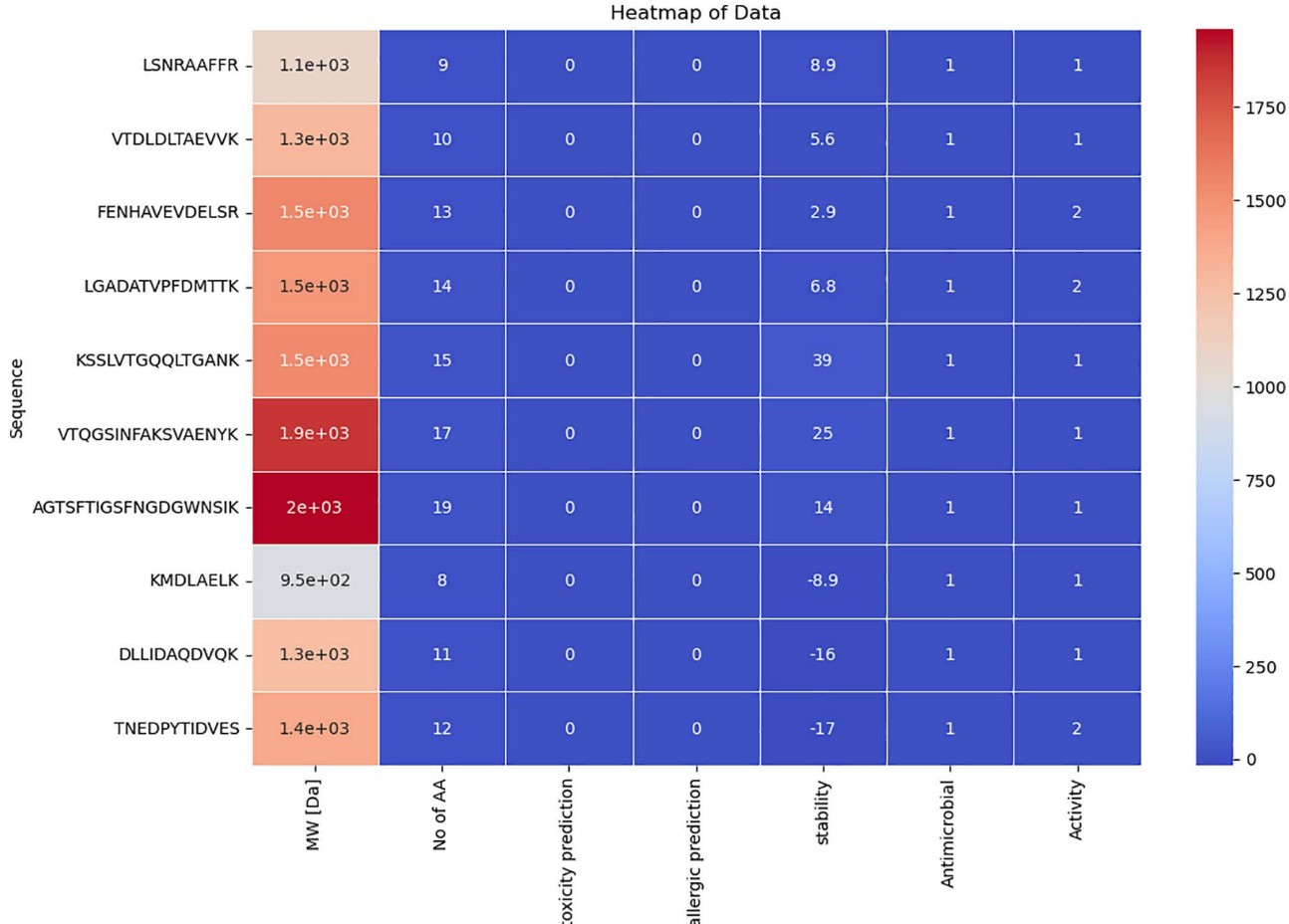

**Fig 2. Peptide profiling of 10 selected bioactive peptides (AMP:1, nontoxin:0, non-allergen:0, activity: ACE inhibitor:1; α-glucosidase inhibitor/ACE inhibitor:2).**

PPI network was created using STRING, resulting in a network comprising 4,007 nodes and 20,292 edges. Within this network, two additional nodes representing IL1RN and FBRS, labelled ENSP00000259206 and ENSP00000348489 respectively, were identified. The PPI networks that were constructed showed significant characteristics. The network displayed a clustering coefficient of 0.366, suggesting a biological network that is well interconnected. The average number of neighbours per node was 15.334, demonstrating the network's deep interconnections. These networks were integrated to incorporate the DN-specific PPI network into the human PPI network, enabling a thorough comprehension of protein interactions in both general cellular processes and DN-specific settings. Moreover, this analysis also helps to understand the complex molecular mechanisms underlying DN, by providing insights into their protein interactions which involves in the disease process, thus, guiding into the development of targeted therapies.

The largest network of primary human PPI is represented in Fig 3, which shows the complex interactions between proteins consisting of 6299 proteins and 1840 interactions, whereas Fig 4 shows the DN-specific PPI network, which is concentrated on genes related to diabetic nephropathy, it consists of 3951 nodes and 20258 interactions. The graphical representations assist deeper investigation and comprehension of protein interactions inside cellular processes and disease mechanisms while also helping to understand the complex structure of network topologies.

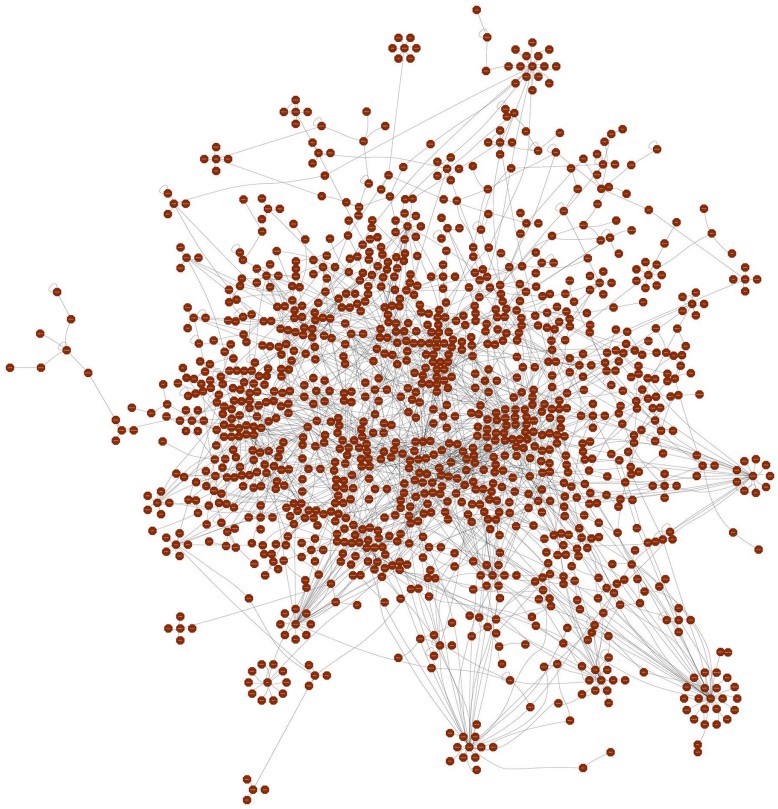

**Fig 3. Primary human PPI network: A comprehensive human protein-protein interaction (PPI) network was constructed by integrating data from three high-throughput databases: HuRI, HINT, and BioGRID.**

Both the human PPI network and the DN-specific PPI network are combined to form the integrated network, which has 3,810 proteins and 39,004 interactions. With a clustering value of 0.270 and an average of 20.385 neighbours per node, it indicates a modest level of interconnectivity (Fig 5). In the combined DN and human PPI network there were increase in interactions.

The MCODE algorithm was used to identify strongly connected areas of the network, which led to the identification of 36 clusters. With an MCODE score of 37.842, cluster 1 stood out as the most significant of these clusters. Cluster 1, also known as the MCODE module, contains 1,080 proteins and 20,416 interactions in total. It has a lot of heterogeneity, as indicated by the score of 0.992. The module also shows an average of 37.807 neighbours per node and a clustering coefficient of 0.409, indicating intense local connectivity. These traits suggest that the proteins in the module are intricately linked together (Fig 6A).

The MCODE module's inclusiveness in the integrated network suggests that the proteins within it are functionally related and probably take involved in the same biological pathways and processes [52]. The protein group represented by the module is cohesive and intricately integrated, and it might be important in a particular biological setting or participate in diabetic nephropathy-related disease pathways.

The investigated outcomes evidently indicate that the proteins in the integrated network are linked functionally and probably take part in similar biological routes and processes. A total of 175 hub nodes were discovered by Cyto-Hubba research (Fig 6B), highlighting their crucial roles in preserving the network's integrity and information flow. Using CytoNCA, additional research unearthed a substantial subnetwork made up of 30 proteins with crucial roles

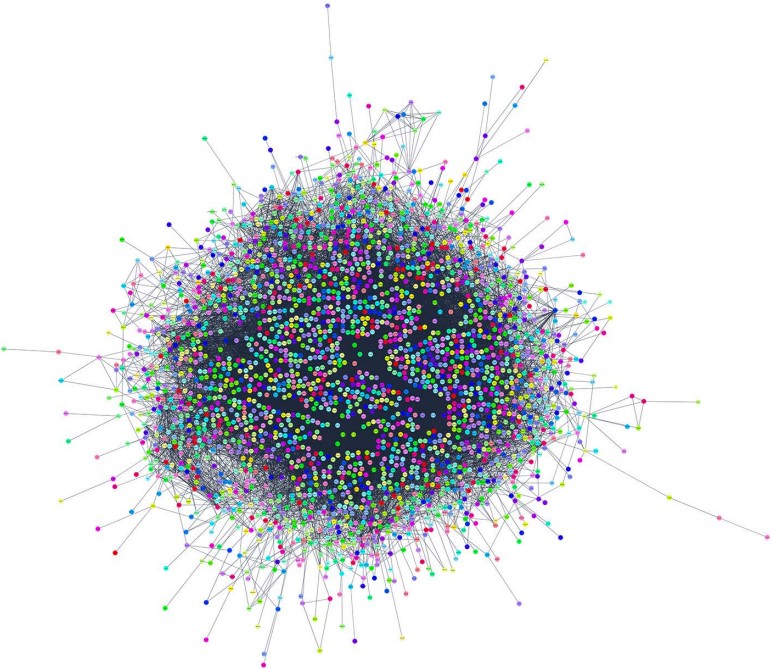

**Fig 4. DN-specific PPI network: protein-protein interaction network of DN associated proteins.**

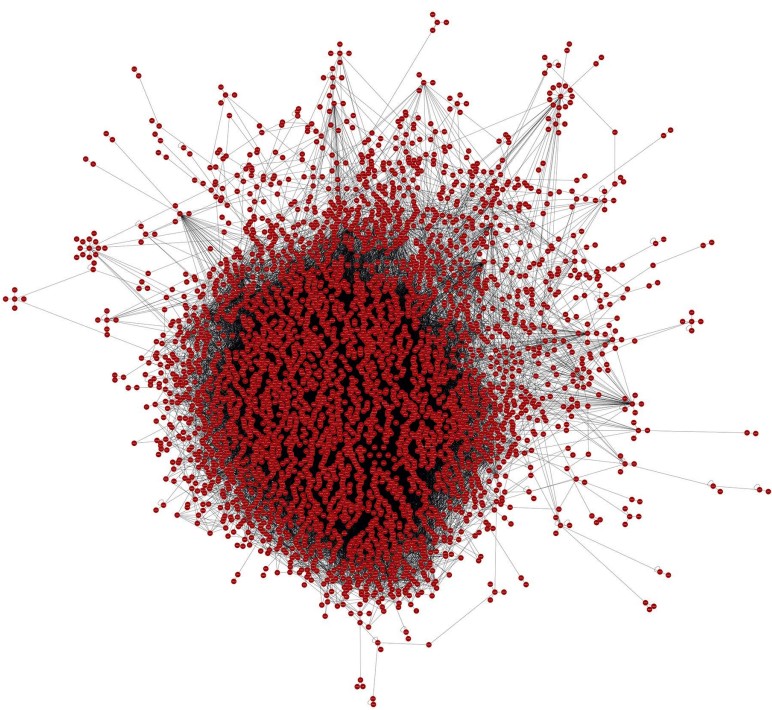

**Fig 5. Integrated network: The integrated protein-protein interaction (PPI) network combines the general human PPI network with the DN-specific PPI network.**

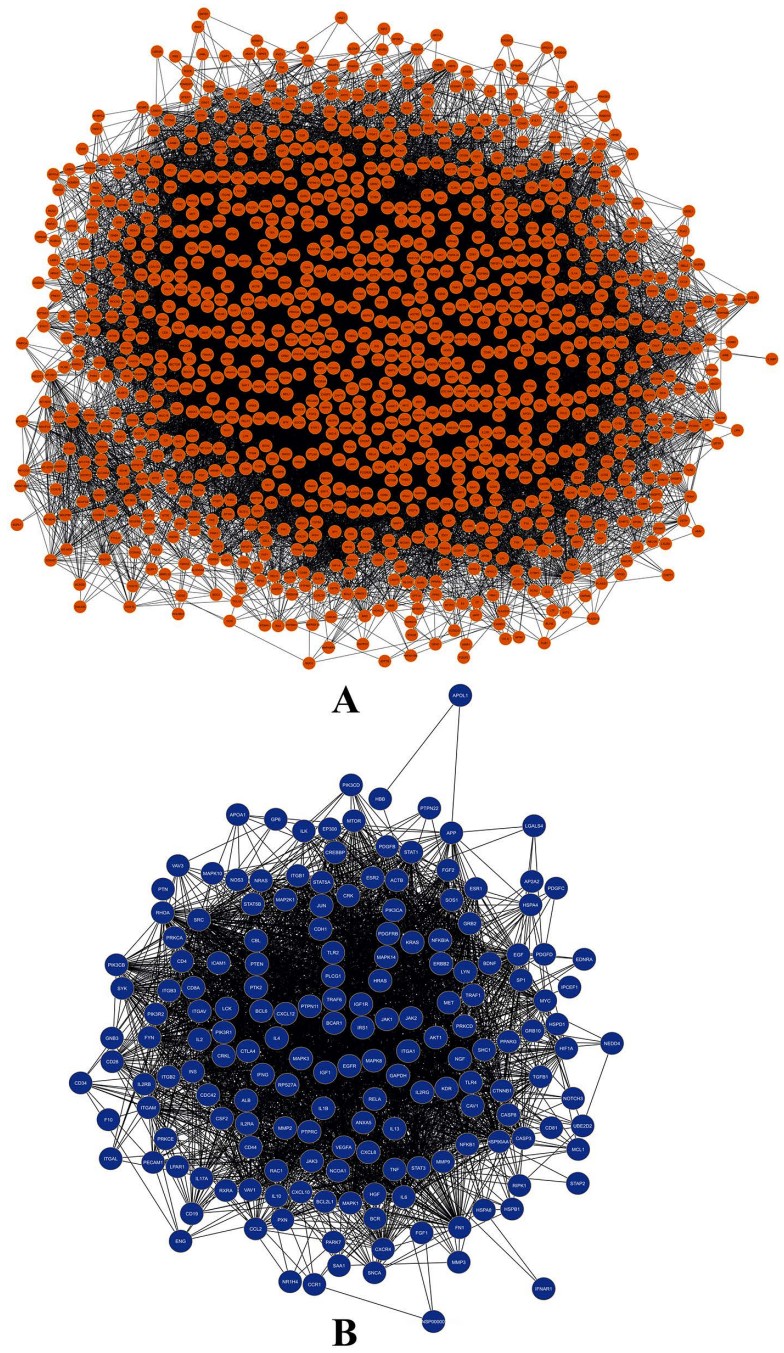

**Fig 6. Computational and topological analysis of integrated network: (A) MCODE analysed and identified module; (B) CytoHubba assessed nodes and network based on 5 algorithms.**

Fig 7. Several important proteins were identified using centrality analysis, which evaluates the significance of nodes in a network's information flow. Among these, the proteins ERK1, AKT1, EGRF, STAT3, SRC, EGF, HRAS, VEGFS, RHOA and KRAS were recognised as important ones that would make good therapeutic targets. These proteins are engaged in the regulation of vital biological processes in addition to playing critical roles in preserving the structural

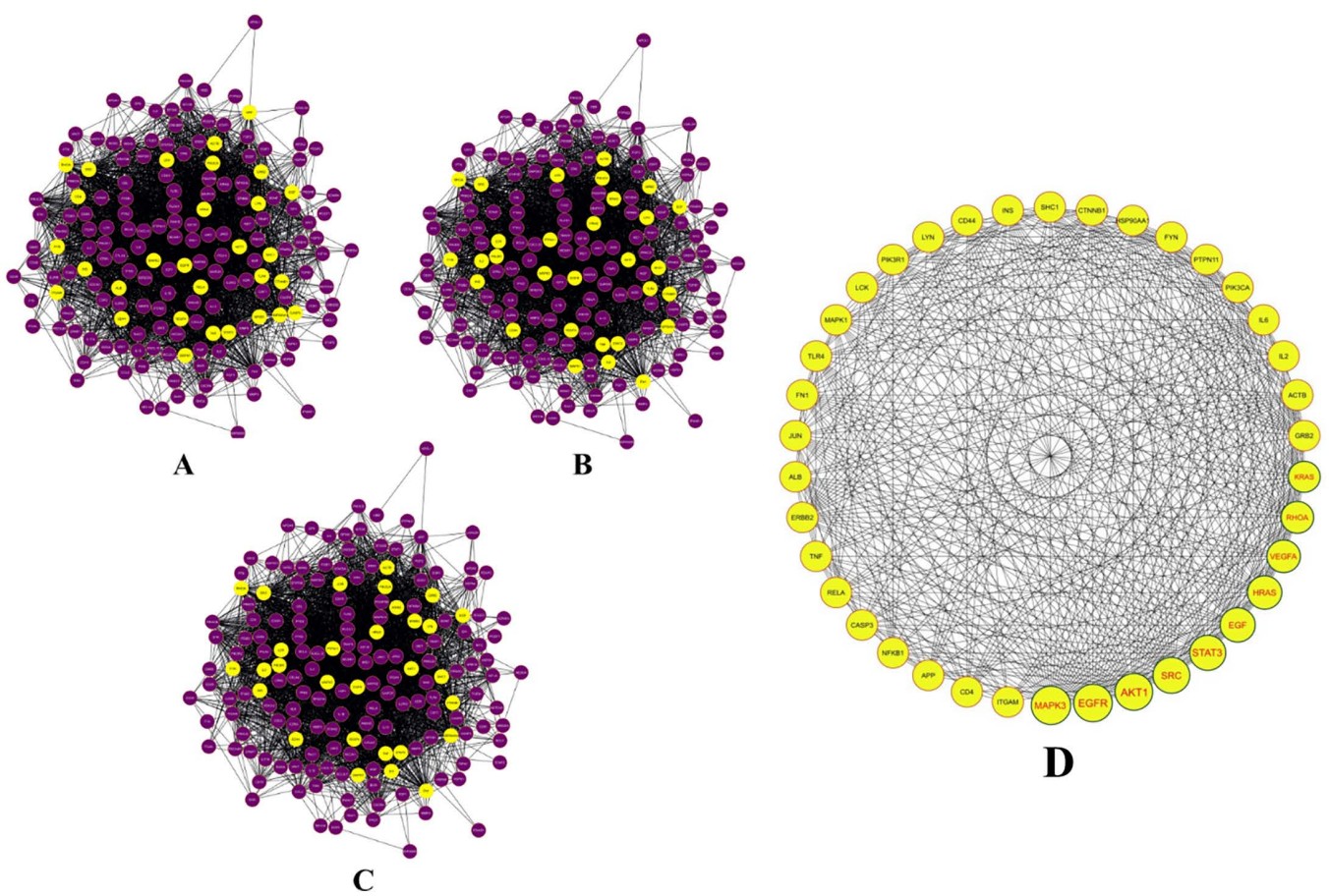

**Fig 7. A) Initial filtration of the network using CytoNCA, highlighting yellow nodes with scores above the median. B)** Second stage of subnetwork filtration via CytoNCA. **C)** Third stage of subnetwork refinement through CytoNCA. **D)** Critical network identified by CytoNCA analysis.

and functional integrity of the network [53,54]. ERK1 (Mitogen-Activated Protein Kinase 3), AKT1 (Protein Kinase B), and STAT3 (Signal Transducer and Activator of Transcription 3) are critical components of various signalling pathways that influence cell growth, survival, and inflammation. Their dysregulation is thought to be responsible for the fibrosis and inflammation that are among the pathological alterations seen in diabetic nephropathy [55–57]. The epidermal growth factor (EGF) and its receptor (EGFR) are key players in cell proliferation, differentiation, and wound healing. The abnormalities in EGF signalling can cause kidney structural and functional abnormalities in diabetic nephropathy [58]. Cell communication and intracellular signalling are both facilitated by SRC. Its function in diabetic nephropathy is connected to the control of kidney-damaging cell adhesion, migration, and matrix remodelling [59]. In diabetic nephropathy, dysregulation of RAS proteins can contribute to aberrant cell proliferation and fibrosis. The proteins ERK1, AKT1, EGRF, STAT3, SRC, EGF, HRAS, VEGFS, RHOA, and KRAS may collectively contribute to the complex molecular landscape of diabetic nephropathy [60]. Thus, targeting these essential proteins offers great potential for the creation of more potent treatments for diabetic nephropathy. It could be feasible to interfere with the illness process and produce therapeutic effects by modifying the activity or function of these proteins. In the search for better treatments for diabetic nephropathy, the identification of these important proteins reveals prospective targets that need further research and offers insightful information for upcoming drug discovery efforts.

Notably, the protein ERK1 consistently showed up as a shared entity across all investigations and had the highest level of interconnectedness within this subnetwork. It's notable that ERK1 constantly placed among the top four outcomes of the CytoNCA analysis. ERK1 was chosen as a strong candidate for additional in-depth examination and focused investigation because of its exceptional prevalence and strategic positioning.

Several experimental studies have demonstrated that the ERK1/2 signaling pathway plays a central role in the pathophysiology of DN. In hyperglycemic conditions, ERK1/2 is activated in renal tissues thus, promoting mesangial cell proliferation, inflammatory cytokine production, and renal fibrosis which is a hallmark of DN progression. ERK1/2 has also been shown to regulate extracellular matrix (ECM) accumulation in glomerular mesangial cells and podocyte dysfunction under diabetic conditions [61–63]. These findings provide strong biological evidence supporting our computational prioritization of ERK1 as a critical therapeutic target in DN.

To further support the findings from the PPI network analysis, GO functional and KEGG pathway enrichment analyses were performed on the critical network generated by CytoNCA, which included top 30 genes. The gene set was overrepresented with multiple functional annotations, as evidenced by GO functional enrichment analysis, resulting in 1000 biological process (BP) terms, 189 cellular component (CC) terms, and 309 molecular functions (MF) terms. The top 10 gene ontology-enriched annotations are represented in Fig 8.

Annotations reveal the functional diversity of the gene set and suggest that it may be involved in multiple cellular functions. The gene set has a highly significant impact on fundamental biological processes crucial for cell functioning crucial, such as the ERK1 cascade gene sets involving sequential molecular events that signal within cells. Protein phosphorylation regulates protein activity by adding phosphate groups. These gene sets positively regulate cell communication and signal transduction [64,65]. Maintaining cellular homeostasis, changing in reaction to changes, and carrying out appropriate physiological responses all depend on the coordinated operation of various biological processes. These activities are crucial for cellular health and function, and their dysregulation can result in a variety of illnesses and disorders [66]. In context to DN Their dysregulation can result in glomerular hypertrophy, fibrosis, inflammation, and reduced renal function that characterise chronic kidney disease. Knowing how these mechanisms interact in complex ways can help clinicians identify possible treatment targets for slowing the course of diabetic nephropathy and enhancing patient outcomes.

Gene products are categorised into several cellular locations and structures by the cellular component ontology. These consist of proteins found at the leading edge of migrating cells, those positioned on the inner side of the plasma membrane, those found in rafts and microdomains, and those linked with vesicles [67]. Additionally, anchoring junction-related intracellular components and perinuclear areas are discovered. This ontology includes membrane protein complexes, proteins involved in postsynaptic specialisation, and proteins found on the plasma membrane's outside. These elements are essential for intracellular organisation, adhesion, transport, and cell signalling. Maintaining cellular integrity, permitting dynamic interactions, and aiding cellular activities in health and illness depends on their proper operation [68,69].

According to their molecular roles, the proteins in concern are essential for protein-protein interactions and cell signalling pathways. They can bind to phosphorylated amino acids, phosphotyrosine residues, and phosphoproteins, showing their participation in complex signalling networks [66]. These proteins also engage in interactions with important molecules such growth factor receptors, phosphatases, kinases, and signalling receptors, highlighting their functions as mediators of critical physiological responses [70]. Their involvement in protein-containing complexes emphasises their importance in a variety of biological processes, including transcription, translation, and enzymatic activity. These molecular activities emphasise the complex network of connections and signalling pathways that these proteins are a part of, supporting the carefully planned operation of the cell.

Our study identified a total of 177 statistically significant pathways, of which the top 30 highly enriched pathways are depicted in dot plots. Pathway analysis elucidates the pathophysiology of diabetic nephropathy (DN) involving interconnected signaling pathways. Prolactin, T cell receptor, ErbB, AGE-RAGE, neurotrophin, EGFR tyrosine kinase resistance inhibitor, CLR, HIF-1, endocrine resistance, Rap1, Ras, and PI3K-Akt signaling pathways are primarily involved in the

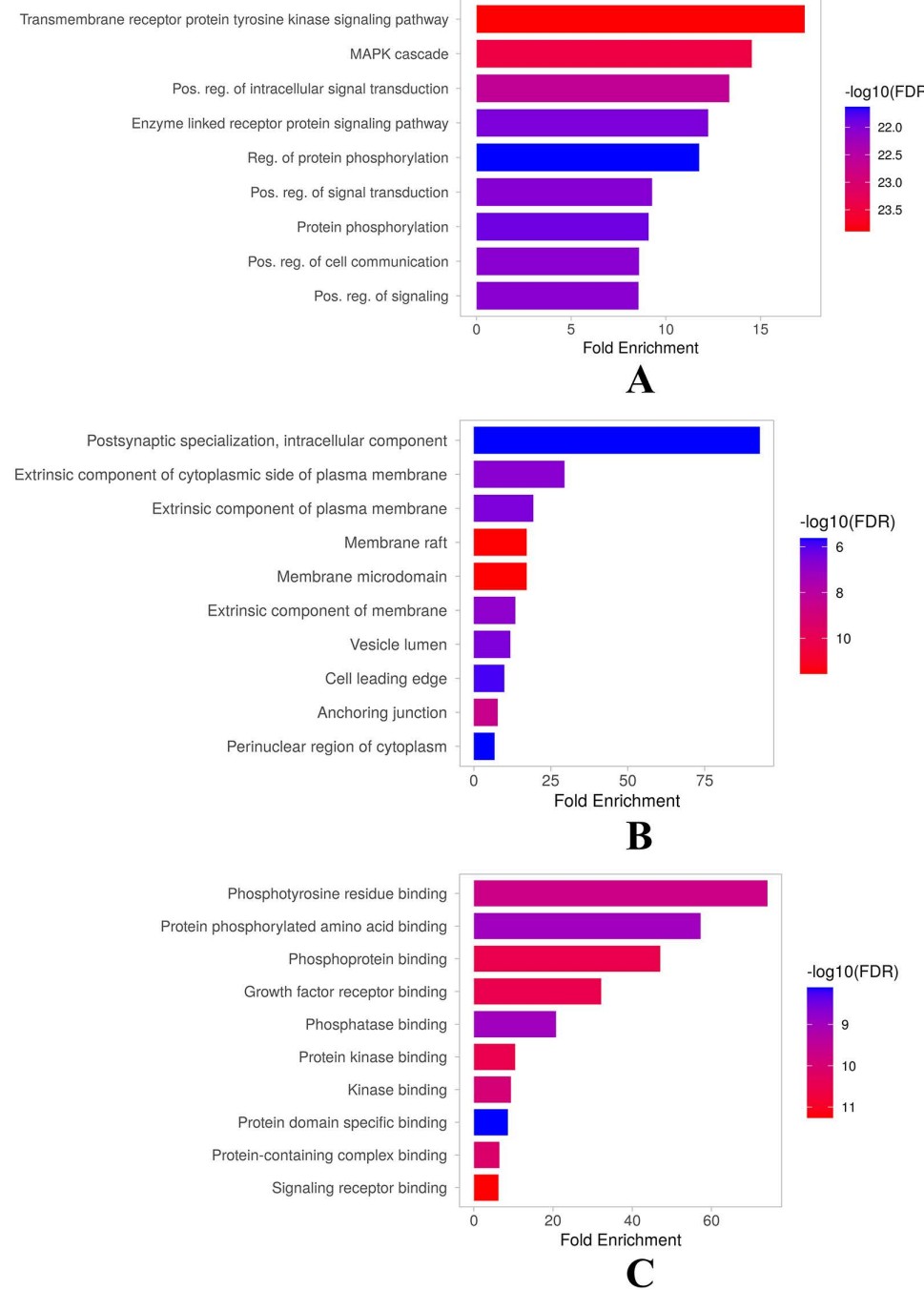

**Fig 8. Results of GO functional enrichment analysis: top 10 highly enriched GO terms (A) BP annotations; (B) CC annotations; (C) MF annotations.**

onset and progression of DN. These genes are also significantly involved in a variety of other diseases and processes, including renal cell carcinoma, regulation of the actin cytoskeleton, focal adhesion, growth hormone synthesis and secretion and action, PD-L1 expression and PD-1 checkpoint in cancer, and Th17 cell differentiation. These pathways are graphed in the dot plot, Fig 9.

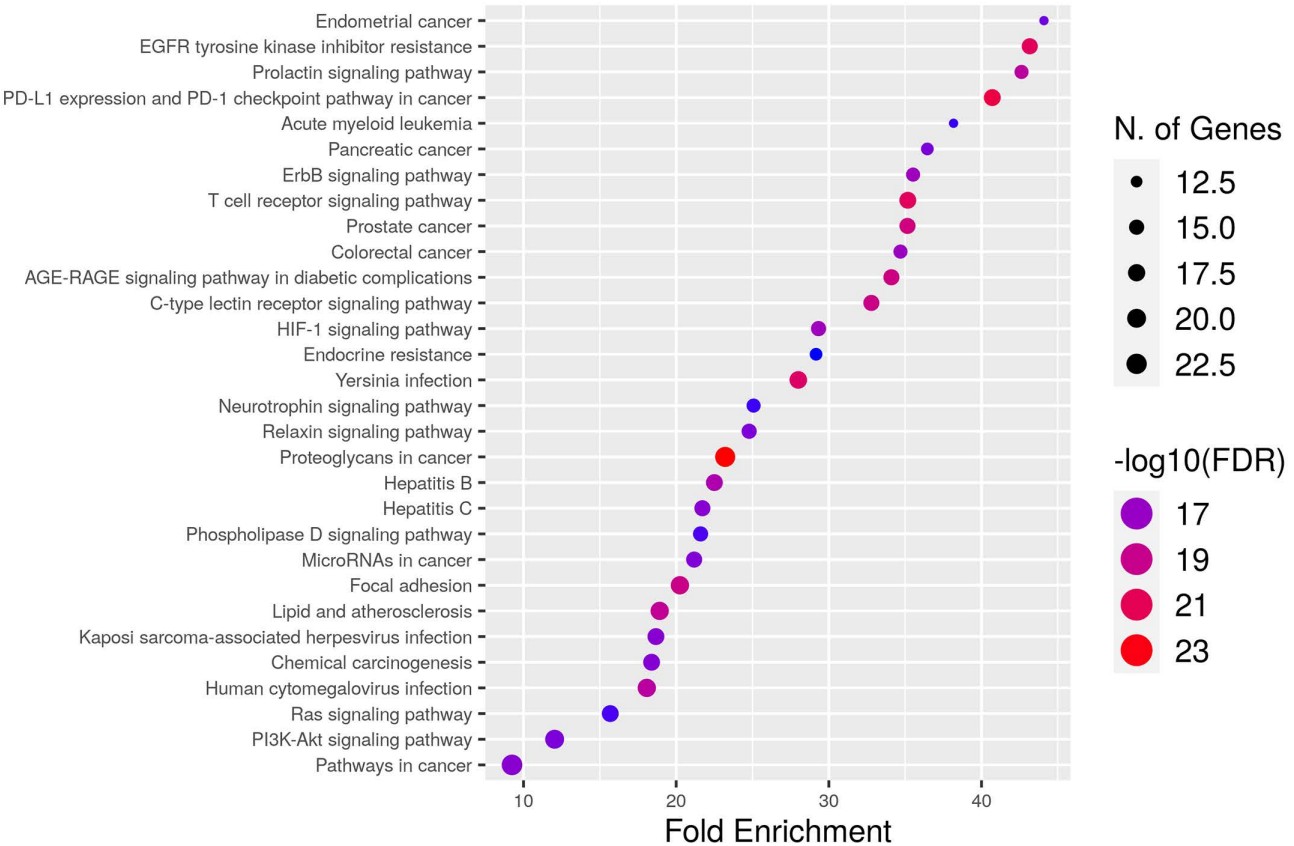

**Fig 9. Results of KEGG enrichment analysis.**

Prolactin regulates a wide range of functions, including lactation, immune system modulation and sodium-potassium transport. Emerging research suggests strong correlation between prolactin signaling pathway and diabetic nephropathy. Prolactin receptors are abundantly expressed in the glomerulus and proximal tubules of the kidney, and individuals with kidney disease often display a sustained increase in prolactin levels [71,72]. Various metalloproteinases and cathepsin D degrade prolactin by proteolysis, resulting in the production of short peptides called vasoinhibins that affect the endothelium. Vasoinhibins are believed to evoke CKD and tubulointerstitial fibrosis through endothelial dysfunction, which in turn causes the kidney to experience oxidative stress and inflammation. Additionally, it promotes the activation of plasminogen activator inhibitor-1 (PAI-1), resulting in the development of fibrosis [73]. PRL bridges the communication between the neuroendocrine system and the immune system. As a cytokine, PRL interacts with monocytes, lymphocytes, NK cells, macrophages and thymic epithelial cells through the prolactin receptor (PRLR). The binding of a ligand to its receptor triggers a cascade of events, including the activation of the P13K/Akt, ERK1 signaling pathways. It regulates the proliferation, survival, selection and activity of T cells and B cells. It also promotes the production of CD40L and IL-6 and proinflammatory cytokines, such as IFN. PRL mediates autoantibodies production and increased cytotoxic activity of T lymphocytes and NK cells in renal lymphoid nodules of systemic lupus erythematosus (SLE) patients [74]. Elevated expression of inflammatory cytokines, macrophage activation and cytotoxic activity are implicated in renal dysfunction and diabetic kidney disease (DKD) [75–77].

Recent research has shown that there is a strong connection between abnormal innate and adaptive immunity, metabolism, and progressive kidney injury. A transcriptomic study of peripheral blood mononuclear cells (PBMCs) and kidney

tissue from diabetic kidney disease (DKD) patients confirmed that the FCER1 pathway is overexpressed in advanced stages of the disease [78]. The study also emphasizes T cell receptor-mediated pathways and increased expression of genes, such as CCR1, TNF and TLRs, which have been implicated with DKD. The production of several inflammatory mediators including IL6, TGF-β, Tryptase,TNF-α and IL1 can be stimulated by elevated intra-renal FcER1 activation in invading mast cells in a diabetic patient. These inflammatory mediators and mast cells stimulate the synthesis of non-fibrillar short-chain type VIII collagen and promote mesangial and extracellular matrix accumulation, which contributes to the development of renal fibrosis [78,79]. At the molecular level following cross-linking of FcER1, downstream signals are transmitted through several pathways, including the ERK1 cascade, the Rho family signaling pathway, and the PI3K signaling pathway. Cross-linking activates Lyn and Syk which initiate signaling complexes of Shc, Grb2, and SOS. These complexes activate Ras proteins, thereby the Raf1-MEK1-ERK1/2 pathway and expression of cytokines and associated genes. Furthermore, the ERK pathway also regulates the release of arachidonic acid (AA) metabolites in human cultured mast cells (HCMCs). In the kidney, high levels of arachidonic acid (AA) metabolites, such as leukotriene B4 (LTB4), thromboxane (Tx), and prostaglandins (PGs), can cause inflammation and damage to the kidney [80]. Downstream activation of FcER1 through Ras and Rho families regulate actin cytoskeleton. Rho-GTPase on the other hand activates FcεRI in RBL cells, affecting actin cytoskeleton organization, calcium ion mobilization and degranulation, and activation of JNK [81].

EGFR (ErbB1), an epidermal growth factor receptor with tyrosine kinase activity, has been extensively investigated for its role in the etiology of DN. EGFR is activated by a variety of ligands, including EGF, TGF-alpha, and HB-EGF [82]. EGFR is expressed in many different types of cells in the kidney, including glomerular endothelial cells, renal epithelial cells, tubular cells, podocytes, medullary interstitial cells and mesangial cells. When high blood sugar levels activate EGFR, it leads to malfunction in a variety of kidney cell types, which can initiate and accelerate kidney damage [83]. It has already been proven that blocking the action of EGFR can reduce the hemodynamic changes that are caused by angiotensin II. EGFR inhibition improves the function of blood vessels in DKD by reducing the thickness of blood vessels through the ERK1/2-ROCK pathway [84]. In individuals with diabetes, EGFR plays an important role in metabolic pathways. EGFR signaling inhibition enhances glucose tolerance and reduces insulin resistance [85]. The AGER-EGFR interaction induces the production of oxidative species such as $H_2O_2$ formation in mesangial cells and in epithelial-like cells of the human fetal kidney. EGFR is also implicated in macrophage infiltration and podocyte damage as well as in inflammatory responses of the kidney in DKD [83].

The cells can contribute to renal tissue damage and inflammation through the production of local cytokines. Primarily, ROR-γt, (TGF)-β, IL-23, IL-6 and IL-1β direct the differentiation of TH17 cells [86]. TGF-β has been demonstrated to contribute to the development of T cells, especially Th17 and Foxp3+Treg cells, in renal inflammatory disorders. Activation of Th17 cells by JAK/STAT signaling leads to the production of the signature cytokine, IL-17. Th17 cells are potentially overexpressed in DKD, resulting in an increased production of IL-17A and IFN-γ [87]. A correlation has been observed between the Th17/Treg ratio and decreased kidney function. This suggests that decreased kidney function may be due to an increase in Th17 cells or a decrease in Treg cells. The IL-17 family is a key mediator for inflammation. The IL-17A triggers inflammatory signaling pathways and regulates T cell viability [88,89]. Additionally, IL-17 also has an effect on the inflammatory activation of macrophages and neutrophils [90]. A study by **Liu H et al., (2013)** [91], investigated the role of ERK1 in the development of Th17 and Treg cells to investigate the pathogenesis of IBD and inflammatory mechanisms. The findings demonstrate that the ERK1 pathway is involved in stimulating CD4+T-cell differentiation into Th17 cells while suppressing Treg-cell development. Blocking the ERK pathway reduced the production of RORγ receptor in response to IL-6, but increased the production of Foxp3 in response to TGF-β. Similar to Treg cells, ERK inhibitor-treated Th17 cells negatively regulate naïve T-cell proliferation and IFN-γ- production *in vitro* by secreting more IL-10 and TGF-β.

A positive feedback loop between chemokines and T cells amplifies the inflammatory response and encourages immunological adaptation. Circulating T lymphocytes are attracted to tissues by chemokines, which promote their infiltration. Additionally, chemokines produced by T cells may modulate the pathophysiological development of renal insufficiency

[87]. The most widely recognized member within the CC chemotactic chemokine family is CCL2, commonly referred to as macrophage chemokine-1 (MCP-1). In DN, CCL2 may serve as a marker for renal inflammation and tubular damage. High glucose promotes CCL2 expression and induces the phosphorylation of IκBα, IκBβ, NF-κB/p65 in podocytes [92]. When AGEs or diabetic circumstances are present, the CXCL9-CXCR3 receptor controls the recruitment of Th1 polarized T cells. In diabetes, AGE binding sites on T cells increase, leading to the synthesis and release of proinflammatory cytokines. In the initial stages of nephropathy, CX3CR1+T cells were found to be elevated and medicate IL-17A production, which is accompanied by renal impairment. CCL5, another β-chemokine, is a chemotactic factor for T cells and is sometimes referred to as RANTES regulated on activation, normal T cell expressed and secreted). In inflammatory kidney disorders, it imposes CD4+T lymphocytes activation and accumulation [93]

All the pathway information emphasises the complex relationships among various signalling pathways and their significance to diabetic nephropathy. A hormone with multiple functions, prolactin, has a direct relationship with diabetic nephropathy. The parts of the kidney that express prolactin receptors are linked to issues with the kidneys. Vasoinhibin is produced when prolactin processing is dysregulated, which contributes to endothelial dysfunction, oxidative stress, inflammation, and fibrosis in the kidney [94]. Complicating factors include the ERK1 cascade, EGFR signalling, and immunological pathways. In EGFR-mediated signalling, cell proliferation, and differentiation, ERK1, a crucial member of the ERK1 cascade, participates. Kidney impairment may be accompanied by abnormal EGFR signalling [95]. Additionally related to the ERK1 and EGFR pathways are the immunological pathways involving T-cell responses, chemokines, and cytokines. Inflammatory and fibrotic processes in diabetic nephropathy can be made worse by dysregulated immune responses [96]. Overall, the crosstalk and interaction across these pathways contribute to the complex pathophysiology of diabetic nephropathy, underscoring the need for thorough understanding and potential therapeutic targeting.

### 3.4. Three-dimensional model generation

Accurate prediction and refinement of peptide structures are crucial for understanding their functional characteristics and potential applications [97]. An overview of the predicted peptide dataset and their validation findings for the predicted structures are given in Table 2 and S1 File **(Supporting information)**. The I-Tasser approach was used to predict the secondary structures of peptides 1–8, while the Pepfold method was used to predict the secondary structures of peptides 9 and 10 because they had fewer than 10 residues. The Ramachandran plot analysis, QMEANDisCo scores, and MolProbity score were used to validate the predicted structures. The Ramachandran plot study evaluates how the peptide's dihedral angles are distributed in favourable, permitted, and unfavourable regions. Using the agreement between anticipated

Table 2. Peptide refinement and validation using I-Tasser and PepFold 3.

| Sl.no | Peptide Name | RMSD of refined model (modrefiner) | Ramachandran plot | | | MolProbity Score | QMEAN-DisCo | C-score (−5–2) (I Tasser) | sOPEP (pep fold3) |
|---|---|---|---|---|---|---|---|---|---|
| | | | Favoured | Allowed | Unfavoured | | | | |
| 1 | VTDLDLTAEVVK | 0.431 | 100.0% | – | – | 0.50 | 0.39 | −0.57 | – |
| 2 | FENHAVEVDELSR | 0.226 | 27.3% | 63.6% | 9.1% | 1.72 | 0.46 | −0.76 | – |
| 3 | LGADATVPFDMTTK | 0.559 | 90.0% | 10.0% | – | 1.46 | 0.27 | −1.55 | – |
| 4 | KSSLVTGQQLTGANK | 0.407 | 100.0% | – | – | 1.20 | 0.38 | −1.69 | – |
| 5 | VTQGSINFAKSVAENYK | 0.424 | 92.9% | – | 7.1% | 0.95 | 0.74 | −0.63 | – |
| 6 | AGTSFTIGSFNGDGWNSIK | 0.114 | 69.2% | 30.8% | – | 2.61 | 0.38 | −1.65 | – |
| 7 | DLLIDAQDVQK | 0.385 | 100.0% | – | – | 1.89 | 0.56 | −0.08 | – |
| 8 | TNEDPYTIDVES | 0.009 | 77.8% | 11.1% | 11.1% | 2.27 | 0.00 | −0.79 | – |
| 9 | LSNRAAFFR | 0.005 | 100.0% | – | – | 1.36 | 0.83 | – | −10.7715 |
| 10 | KMDLAELK | 0.001 | 100.00% | – | – | 0.50 | 0.71 | – | −12.0533 |

and experimental structures, the QMEANDisCo score assesses the quality of the predicted structures [98] These results offer valuable insights into the reliability and suitability of the modelled peptide structures for further studies and applications, which will benefit future research and applications (Fig 10).

RMSD scores, which measure structural deviation from a baseline model, have shown varying degrees of concordance between the refined peptide mode. In particular, the "TNEDPYTIDVES" and "LSNRAAFFR" peptides had unusually low RMSD scores of 0.009 and 0.005, respectively, indicating elevated levels. The structural models of these peptides are similar to those of their experimental counterparts, suggesting that they can be used to construct potential functional models.

Additionally, the distribution of dihedral angles within the revised peptide structures was evaluated using the Ramachandran plot approach. The Ramachandran plot showed a high percentage of favoured residues for the peptides VTDLD-LTAEVVK, KSSLVTGQQLTGANK, DLLIDAQDVQK, LSNRAAFFR, and KMDLAELK, ranging from 90.0% to 100.0%. This finding suggests that these peptides are in a desirable conformational arrangement, further demonstrating their structural

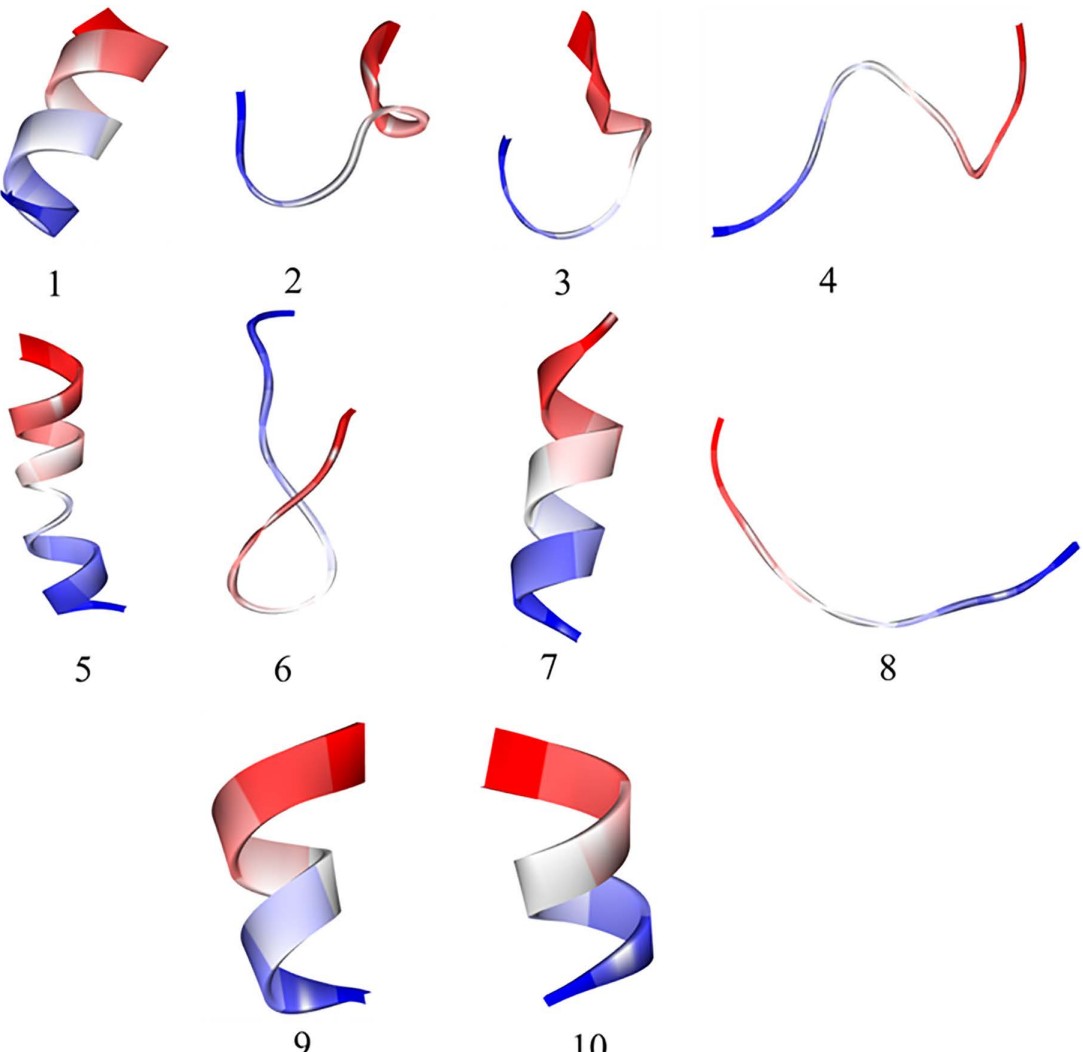

**Fig 10. 3D-built models of bioactive peptides mentioned in Table 2 and S1 File (Supporting information).**

stability and appropriateness for further research [43,99]. Further details were provided by the MolProbity ratings, which rate the refined models' stereochemistry and general quality. Notably, the peptides KMDLAELK and VTDLDLTAEVVK had low MolProbity scores of 0.50, indicating outstanding structural integrity and advantageous stereochemical characteristics. The precise folding patterns and conformations of these peptides increase their potential for functional investigation and use across a range of scientific research [99]. The reliability of the refined peptide models was further evaluated using the QMEANDisCo scores, which took into account the consistency between predicted and experimental structures. With a QMEANDisCo score of 0.83, the peptide LSNRAAFFR demonstrated the best model quality and dependability.

Overall, several possibilities for subsequent study and potential uses were revealed by the comparative analysis and quality assessment of the revised peptide structures. Due to their low RMSD scores, large proportion of preferred residues in the Ramachandran plot, and superior QMEANDisCo scores, the peptides "TNEDPYTIDVES" and "LSNRAAFFR" stood out as highly promising choices. The structural models provided by these peptides are trustworthy and precise, paving the way for in-depth research into their functional characteristics and possible uses in areas including medication development and biological research. The physicochemical properties of both the selected peptides have been given in Table 3.

## 3.5. Molecular docking and interaction analysis

The HADDOCK server was used to perform molecular docking after generating the molecular models in an effort to better comprehend the interactions that bind peptides to the ERK1 target. The computational docking data (Table 4 **and Supporting information** S1 File) included a number of factors, such as Dock score, Z-score, RMSD, G (kcal mol-1), and Kd values, which revealed data about the expected binding affinity and properties of the peptides.

The Dock score reflects the overall binding affinity between a peptide and its target, with a lower score indicating a stronger binding interaction [100–102]. In the study, LSNRAAFFR exhibited a dock score of −90.2 ± 2.0, while TNEDPYTIDVES displayed a dock score of −110.6 ± 4.3. In addition, the Z-score indicates the relative binding affinity of a peptide with respect to other molecules, with lower values representing a higher affinity. LSNRAAFFR achieved a Z-score of −1.7, whereas TNEDPYTIDVES attained a Z-score of −1.5. The RMSD, which quantifies the structural alignment quality by measuring the average distance between the docked peptide and the reference or experimental structure, was assessed as well. LSNRAAFFR yielded an RMSD of 3.2 ± 0.1, whereas TNEDPYTIDVES exhibited an RMSD of 2.7 ± 0.0. These values contribute to understanding the accuracy of the structural alignment.

The Van der Waals energy term, which accounts for the attractive and repulsive forces that exist between the atoms involved in van der Waals interactions, contributes to the stability of the peptide-receptor complex. While TNEDPYTIDVES had a Van der Waals energy of −63.3 ± 3.8, LSNRAAFFR showed a Van der Waals energy of −54.0 ± 3.5. The overall stability and binding affinity are also influenced by electrostatic energy, which refers to the electrostatic interactions between charged atoms in the complex. While TNEDPYTIDVES displayed an electrostatic energy of −294.7 ± 7.8, LSNRAAFFR displayed an electrostatic energy of −108.5 ± 29.7.

ΔG represents the free energy change upon binding of the peptide to its target and provides an estimation of the thermodynamic stability of the complex. LSNRAAFFR exhibited a ΔG of −8.8 kcal mol-1, while TNEDPYTIDVES displayed a ΔG of −9.9 kcal mol-1. Additionally, the Kd (Dissociation Constant) represents the equilibrium dissociation constant and indicates the peptide concentration required for 50% binding to the target at a specified temperature. the results estimated that LSNRAAFFR to have a Kd of 3.2e-07 M, while TNEDPYTIDVES demonstrated a Kd of 5e-08 M. Nevertheless, it's important to understand that these values are predicted and they require experimental validation using various techniques like surface plasmon resonance to confirm their accuracy.

Furthermore, these results offer valuable insights into the binding characteristics and potential affinity of the peptides to their respective targets. The lower dock score, Z-score, RMSD values, and more negative ΔG, Van der Waals, and electrostatic energy values associated with TNEDPYTIDVES suggest a stronger binding and more stable interaction compared to LSNRAAFFR. However, further analysis and experimental validation are crucial to confirm and interpret the

**Table 3. Physico-chemical properties and structural details of the selected peptides detected in the study.**

| Particulars | LSNRAAFFR | TNEDPYTIDVES |
|---|---|---|
| **Amino Acid Composition** | | |
| Ala (A) | 22.2% | 0.0% |
| Arg (R) | 22.2% | 0.0% |
| Asn (N) | 11.1% | 8.3% |
| Asp (D) | 0.0% | 16.7% |
| Cys (C) | 0.0% | 0.0% |
| Gln (Q) | 0.0% | 0.0% |
| Glu (E) | 0.0% | 16.7% |
| Gly (G) | 0.0% | 0.0% |
| His (H) | 0.0% | 0.0% |
| Ile (I) | 0.0% | 8.3% |
| Leu (L) | 11.1% | 0.0% |
| Lys (K) | 0.0% | 0.0% |
| Met (M) | 0.0% | 0.0% |
| Phe (F) | 22.2% | 0.0% |
| Pro (P) | 0.0% | 8.3% |
| Ser (S) | 11.1% | 8.3% |
| Thr (T) | 0.0% | 16.7% |
| Trp (W) | 0.0% | 0.0% |
| Tyr (Y) | 0.0% | 8.3% |
| Val (V) | 0.0% | 8.3% |
| Pyl (O) | 0.0% | 0.0% |
| Sec (U) | 0.0% | 0.0% |
| **Atomic Composition** | | |
| Carbon (C) | 49 | 58 |
| Hydrogen (H) | 76 | 87 |
| Nitrogen (N) | 16 | 13 |
| Oxygen (O) | 12 | 26 |
| Sulfur (S) | 0 | 0 |
| **No. of AA** | 9 | 12 |
| **Formula** | C49H76N16O12 | C58H87N13O26 |
| **GRAVY** | −0.033 | −1.158 |
| **Theoretical pI** | 12 | 3.43 |
| **Aliphatic index** | 65.56 | 56.67 |
| **Instability index** | 8.89 | 21.67 |
| **Molecular weight** | 1382.40 | 1081.24 |

**Table 4. The interaction result of protein and peptide complex.**

| Sl.no | Peptide Name | Dock score | Z-score | RMSD | Van der Waals energy | Electrostatic energy | ΔG (kcal mol$^{-1}$) | $K_d$ (M) at°C |
|---|---|---|---|---|---|---|---|---|
| 1 | LSNRAAFFR | −90.2±2.0 | −1.7 | 3.2±0.1 | −54.0±3.5 | −108.5±29.7 | −8.8 | 3.2e-07 |
| 2 | TNEDPYTIDVES | −110.6±4.3 | −1.5 | 2.7±0.0 | −63.3±3.8 | −294.7±7.8 | −9.9 | 5e-08 |

significance of these docking results. Further, molecular dynamic simulations were used to examine the dynamic expression for the docked complexes.

The interaction study between two peptides, LSNRAAFFR and TNEDPYTIDVES, and specific residues within the ERK1 protein is represented in Fig 11. The binding interactions depict that both peptides occupy the exact binding region of the co-crystallized ligand SCH772984, previously bound to the structure of human ERK1 (PDB ID: 4QTB) [36].

With LSNRAAFFR, the peptide interacts electrostatically and through hydrogen bonds with the protein. Notably, the arginine (peptide ARG4) of the peptide and the glutamate (GLU50) of the protein form an electrostatically driven hydrogen bond, which ensures the stability of the complex. Signal transmission may be impacted by an interaction between the nitrogen of peptide ARG9 and the oxygen of GLU203 in the protein. The peptide ARG4 and the protein ALA52 form hydrogen bonds, which may have a function in molecular recognition or structural stability. The peptide's PHE7 and ARG84 form many hydrogen bonds that improve the structure's overall stability. Additional hydrogen bonds have an impact on the

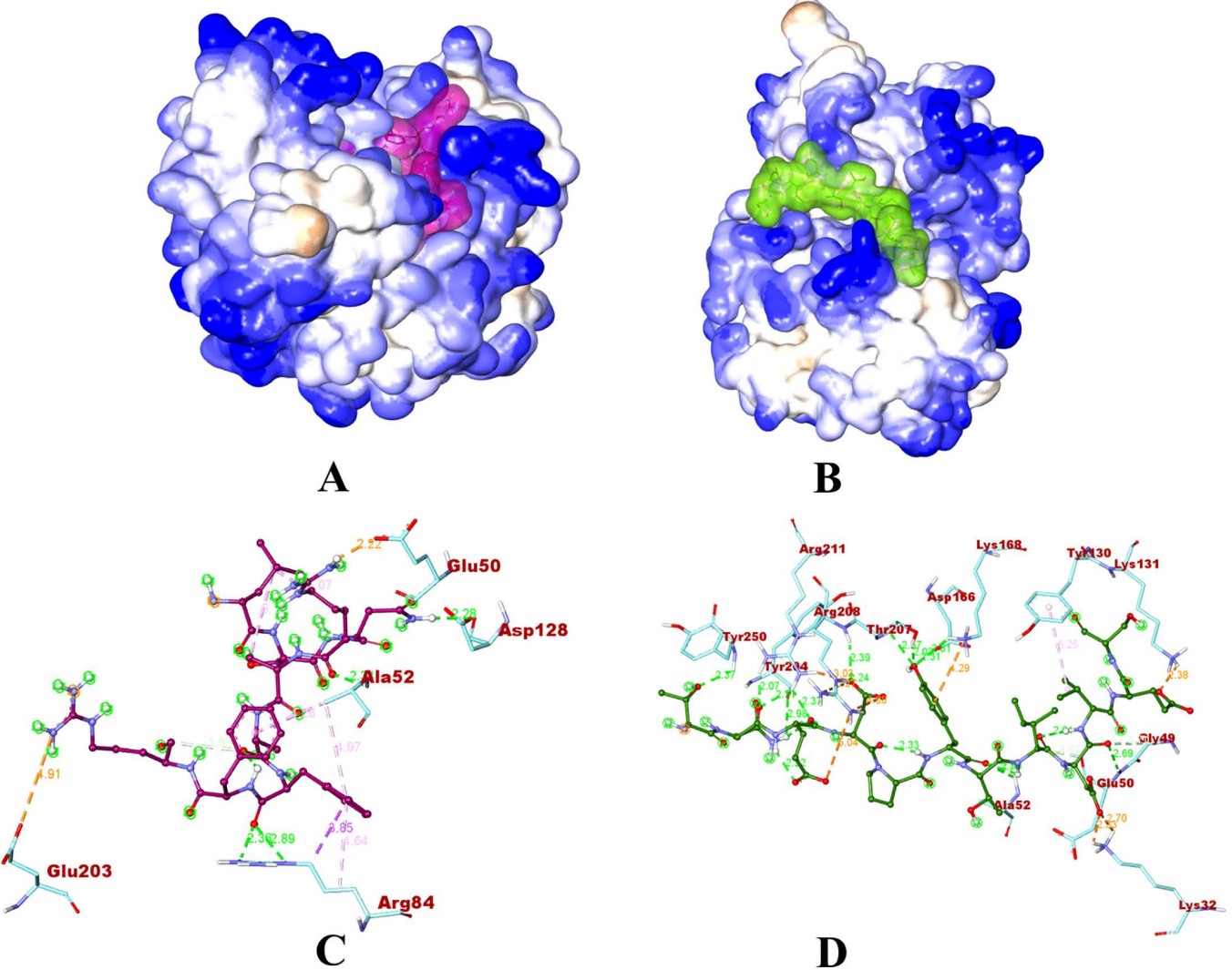

**Fig 11. A and B depict the molecular surface of the ERK1 protein in the presence of two peptides: LSNRAAFFR (purple) and TNEDPYTIDVES (green).** On the other hand, **C** and **D** illustrate the 3D interactions and associated distances between these peptides and the protein.

internal stability and local structure. Stability is also influenced by hydrophobic interactions between different peptide and protein residues. Together, these interactions mould the peptide-protein complex and provide insight into its structural and functional characteristics.

The interactions between the ERK1 and peptide TNEDPYTIDVES are also varied. Major roles are played by hydrogen bonds, electrostatic interactions, and hydrophobic contacts. Hydrogen bond-electrostatic interactions between ASP9 of the peptide and LYS32 of the protein operate as anchors for the complex, showing complex stabilisation. The peptide LYS131 further forms a hydrogen bond with the protein GLU11, highlighting electrostatic contributions. The protein aspartate (ASP4) and the peptide arginine (ARG211) exhibit electrostatic interactions. The contacts between the protein ASP4, protein GLU3 and the peptide ARG208 reveal additional electrostatic interactions that may have molecular recognition roles. The peptide's TYR6 hydrogen bonds have a variety of functional effects. Small conformational alterations are augmented by hydrophobic contacts like the one between TYR130 and VAL10. The knowledge of molecular recognition, structural stability, and signalling mechanisms in complex biological contexts is aided by these interactions taken as a whole.

We also performed hydrogen and hydrophobic bond mapping for the residues bound with the peptides. This was done to identify key residues which could be targeted to design specific-binding ligands in the future. Residues GLU50 and ARG 211 are bound twice with the peptide LSNRAAFFR. Similarly, peptide TNEDPYTIDVES was bound twice with ARG84 with hydrogen bonds (Fig 12). During hydrophobic bond mapping, ALA52 was bound twice with LSNRAAFFR, and TYR130 was the only residue of the ERK1 to be bound with TNEDPYTIDVES (Fig 13).

The peptides were bound to the adenine-mimicking binding pocket, surrounded by the P-loop, ATP/Mg$^{2+}$-coordinating DFG motif (which regulates the catalysis of protein kinase and contributes to ATP binding), αC, and αD domains. Since the binding of the peptides is in the ATP binding site, the ATP/Mg$^{2+}$ coordination may be interfered with by the same [36]. This binding could alter the catalysis of protein kinase and the biological activity (phosphorylation) could be inhibited. Therefore, these interactions play a crucial role in shaping molecular behaviour and potential functional outcomes. They provide insight into the complicated interactions between peptides and proteins, influencing their structural complexity and biological functions. The key residues involved in peptide binding include GLU50, ARG211, ARG84, ALA52, and TYR130, suggesting they may be critical to anchoring ligands in the adenine-mimicking site, thus suggesting that the peptides may interfere with ATP binding by competing for hydrogen bonds or creating steric hindrance. Notably both peptides

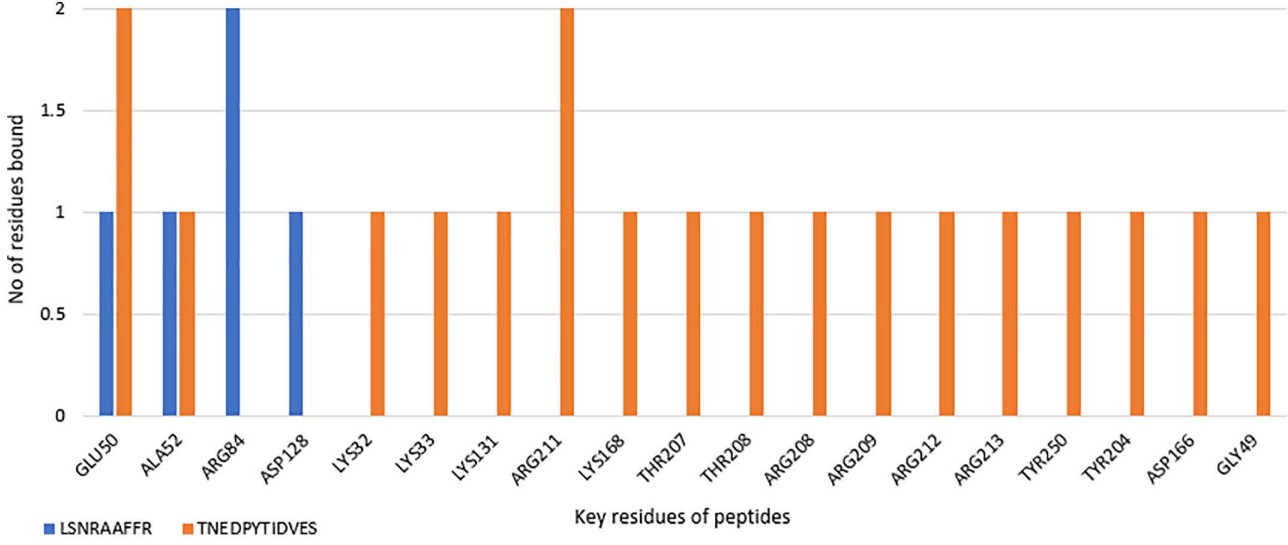

**Fig 12. Hydrogen bond mapping of bioactive peptides docked with ERK1 protein.**

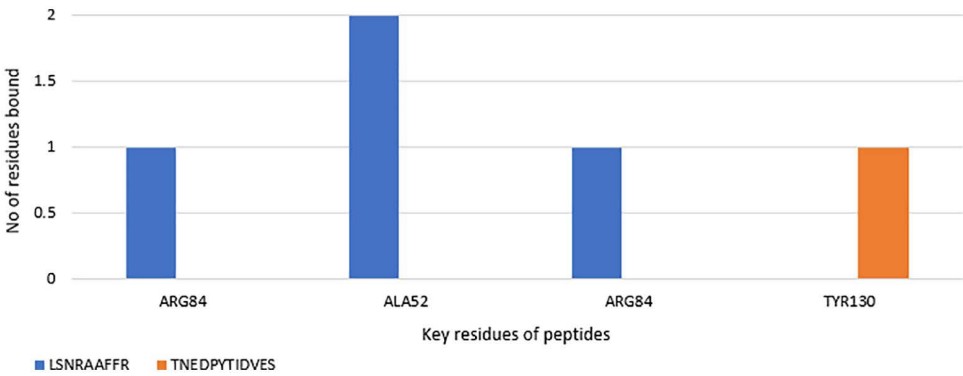

**Fig 13. Hydrophobic bond mapping of bioactive peptides docked with ERK1 protein.**

were found to occupy the same ATP binding cleft as the known ERK1 inhibitor SCH772984 (PDB: 4QTB) thus indicating potential competitive inhibition. This structural overlap strengthens the hypothesis that these peptides could be functionally similarly to small molecule inhibitors by competing with ATP for binding, thereby blocking ERK1 kinase activity or by hinder ERK1's ability to phosphorylate downstream targets.

Further studies are needed to validate this hypothesis and elucidate the precise mechanism of inhibition. A thorough review of each residue's involvement and the distances between them is presented in the detailed interaction table in S1 File **(Supporting information)**.

### 3.6. Molecular dynamics simulations

A powerful method for examining the stable binding modes of peptides to proteins and learning about changes in their conformational stability is molecular dynamics simulation. The trajectories produced by the simulated method provide comprehensive data about the molecular interactions and stability of complexes. The Root Mean Square Deviation (RMSD), Root Mean Square Fluctuation (RMSF), Radius of Gyration (RG), Solvent Accessible Surface Area (SASA), and hydrogen bonds were used as important metrics in this investigation. These variables played a crucial role in determining how stable the docked complexes were.

In regard to RMSD values, the stability of the LSNRAAFFR-ERK1 and TNEDPYTIDVES-ERK1 complexes was meticulously studied. The RMSD calculates the structural variation of the complex over time. It is noticeable from Fig 14A that both complexes stabilised after 75 nanoseconds (ns) at a size of 0.25 nanometers (nm). The protein also fluctuated very little during the course of the simulation. Lower RMSD levels typically denote more stability. The TNEDPYTIDVES-ERK1 complex showed higher stability with fewer variations over the whole molecular dynamics (MD) run, despite the fact that the average RMSD values for the two peptides were similar. Analysing the RMSF plot (Fig 14B) facilitated to assess the flexibility of the protein-ligand complexes. The variation of individual residues over time is calculated by RMSF. The plot demonstrates that, in contrast to the centre sections, where there was little fluctuation, the complexes and the protein showed greater fluctuation near the terminal ends of the complex models. According to this, these complexes may be structurally flexible, especially near their ends. As observed in Fig 14C, the Radius of Gyration (Rg) value was used to gauge how compressed the molecules had become over time. The average Rg value for both complexes remained about 2.15 nm during the course of the simulation, showing that both complexes acquired stability. The protein, on the other hand, showed a marginally larger variation, with an Rg value of roughly 2.22 nm. To gain insight into conformational changes during complex formation and to learn more about the region surrounding the hydrophobic core generated in the complexes, the solvent-accessible surface area (SASA) was examined. The SASA plot, shown in Fig 14D, showed that

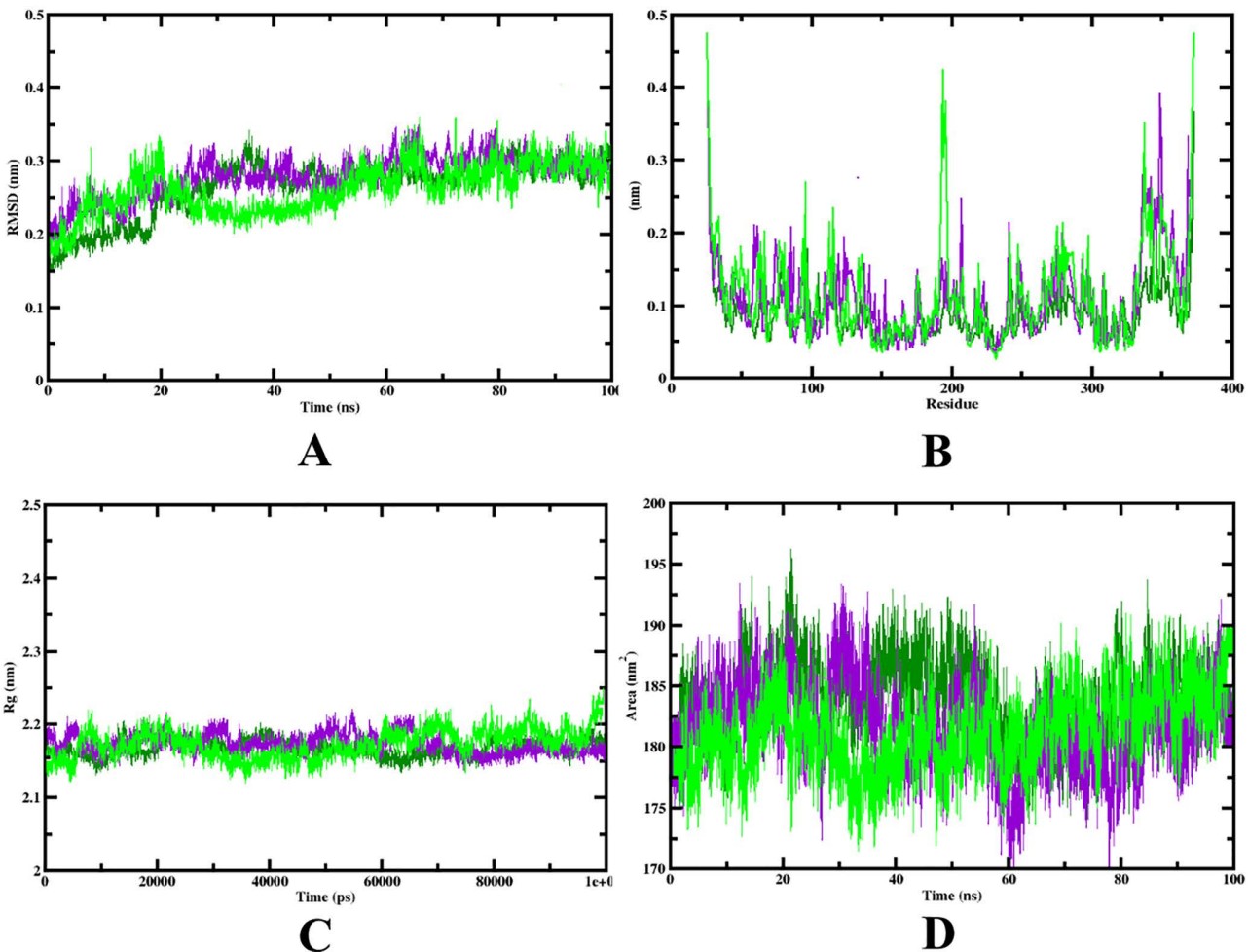

**Fig 14. MD trajectory values of peptides LSNRAAFFR (purple) and TNEDPYTIDVES (dark green) complexed with ERK1 (light green) analysis of (A) RMSD, (B) RMSF, (C) Rg and (D) SASA.**

the complexes stabilised at about 80 ns. The MD trajectory values of the protein-peptide complexes have been given in Table 5.

While longer MD simulation approaching the μs scale are becoming standard for larger biomolecular systems, we selected a 100 ns duration in this study due to its small size of the peptides that is 9–12 amino acids and the nature of the interactions with ERK1. For instance, in a study by **Mu et al., 2005** performed a 100 ns MD simulation of penta-alanine in explicit water to investigate its folding and unfolding dynamics, demonstrating that such timescales can effectively capture the conformational landscape of small peptides [103,104]. Thus, showing that the peptide–protein systems of this scale typically reach equilibrium within 50–100 ns. In our analysis, the key structural metrics such as RMSD, Rg, SASA, and hydrogen bonding converged within the simulated timeframe, indicating that the system had achieved a stable state and the chosen simulation length was adequate for capturing the binding dynamics.

In the final analysis, it appears that the "TNEDPYTIDVES-ERK1" complex had greater stability and maintained a more stable binding mode throughout the molecular dynamic simulation based on the RMSD, RMSF, and Rg studies. The binding mode of the peptide confers stability inside the adenine-mimicking pocket of the ERK1, which is the site of ATP

**Table 5. MD values of peptides LSNRAAFFR and TNEDPYTIDVES complexed with ERK1.**

| MD trajectory values | Apo-protein | Protein-LSNRAAFFR complex | Protein-TNEDPYTIDVES complex |
|---|---|---|---|
| RMSD | 0.25-0.26 nm | 0.25 nm | 0.25 nm |
| Rg | ~ 2.22 nm | 2.15 nm | 2.15 nm |
| SASA | 175-183 nm$^2$ | 175-183 nm$^2$ | 175-183 nm$^2$ |

binding [36]. Since the peptide molecule is found to be stable inside the binding pocket without showing many fluctuations in RMSD, RMSF, Rg, and SASA graphs, it becomes evident that the peptide molecule is binding efficiently and can induce biological activity. The overall stability and behaviour of these complexes can be influenced by a variety of circumstances, therefore more analyses and background information may be required to fully comprehend their stability.

While this study provides valuable preliminary insights into the potential of *Lactobacillus brevis* RAMULAB49-derived peptides to target ERK1 in diabetic nephropathy, it is important to emphasize that the findings are based solely on computational predictions. *In silico* approaches, including molecular docking and molecular dynamics simulations, serve as effective tools for prioritizing therapeutic targets and identifying promising bioactive candidates. However, they do not substitute for empirical validation. The current results are intended to inform and guide subsequent experimental investigations rather than provide conclusive evidence of therapeutic efficacy. Accordingly, the authors plan to conduct detailed *in vitro* kinase assays and *in vivo* studies to evaluate the interaction between ERK1 and the identified lead peptide, and to assess its biological outcomes in relevant models. These follow-up studies will be critical in validating the therapeutic potential and biological relevance of ERK1 inhibition in the context of DN.

## 4. Conclusion

The study highlights the *in-silico* binding of 2 selected bioactive peptides from *Levilactobacillus brevis* RAMULAB49 protein hydrolysate to the human ERK1 protein ATP binding site. The study also focusses on the identification of human ERK1 as the potential target for DN through network pharmacology approaches as well as profiling and 3D modelling of the functional bioactive peptides from the *L. brevis* RAMULAB49 protein hydrolysate, hence highlights the modelling of both target and therapeutic compounds.

The PPI network built using the genes related to DN revealed important proteins and functional components important to the pathogenesis of DN. Among the identified hub proteins, ERK1 was prioritized due to its well-established involvement in mesangial cell proliferation, inflammation, and fibrosis, which are hallmarks of DN progression. Gene ontology and KEGG pathway enrichment analyses further highlights the complex, interconnected biological pathways involved in DN pathogenesis.

During the nLC-ESI MS/MS analysis of the protein hydrolysate, several peptides were identified, out of which only 10 were selected for 3D modelling based on their predicted bioactivity, allergenicity, toxicity and potential for ERK1 interaction. These were docked to the ATP binding site to adenine mimicking the binding site of the human ERK1 protein. Peptide identification was performed with stringent validation parameters, including a false discovery rate below 1%, and peptide scores above accepted thresholds to minimize false positives. Future work will explore peptide quantification through label-free quantification to assess relative abundance.

In interaction study, out of 10 only 2 peptides were found to bind effectively with the key residues of the target protein were found to be stable during the MD simulation, suggesting efficient and specific binding to the ERK1 active site could induce biological activity during *in vitro* studies since they are structurally stable inside the binding pocket. Out of the 2 peptides docked and simulated, TNEDPYTIDVES showed superior interactions with ERK1. The physicochemical and 3D modeling validation analyses also depict their efficient structural stability. This comprehensive approach lays a strong foundation for future *in vitro* validation, in vivo efficacy assessment, and potential therapeutic development of these peptides for the treatment of diabetic nephropathy.

## Supporting information

**S1 File. Supplementary tables on Pep ID, struct, dock. MD.**
(DOCX)

**S2 File. Profiling of peptides.**
(DOCX)

## Acknowledgments

All the authors thank JSS Academy of Higher Education and Research (Mysore, Karnataka, India) for their kind support and encouragement.

## Author contributions

**Conceptualization:** Ramith Ramu.

**Data curation:** Ramith Ramu, Ashwini P.

**Formal analysis:** Reshma Mary Martiz, Shashank M Patil, Kefeng Li, Xijun Tang.

**Funding acquisition:** Mohammad Raish, Alina Sionkowska.

**Investigation:** Reshma Mary Martiz, Hemalatha Nambisan, Mahesh B.

**Methodology:** Reshma Mary Martiz, Ameer Suhail, Piotr Bełdowski.

**Project administration:** Ramith Ramu, Hemalatha Nambisan, Ashwini P, Alina Sionkowska.

**Resources:** Ramith Ramu, Hemalatha Nambisan, Mahesh B, Maciej Przybyłek.

**Software:** Ameer Suhail, Piotr Bełdowski, Kefeng Li.

**Supervision:** Hemalatha Nambisan, Ameer Suhail, Ashwini P, Mahesh B, Maciej Przybyłek, Piotr Bełdowski, Alina Sionkowska, Kefeng Li, Xijun Tang.

**Validation:** Mohammad Raish, Shashank M Patil, Ashwini P, Piotr Bełdowski, Kefeng Li.

**Visualization:** Mohammad Raish, Maciej Przybyłek, Alina Sionkowska, Xijun Tang.

**Writing – original draft:** Reshma Mary Martiz, Ramith Ramu.

**Writing – review & editing:** Hemalatha Nambisan, Ameer Suhail, Mohammad Raish, Shashank M Patil, Ashwini P, Mahesh B, Maciej Przybyłek, Piotr Bełdowski, Alina Sionkowska, Kefeng Li, Xijun Tang.

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
