## [Decision Letter · Decision Letter 0]

8 Apr 2025

Dear Dr. Ramu,

Thank you for submitting your manuscript to PLOS ONE. After careful consideration, we feel that it has merit but does not fully meet PLOS ONE’s publication criteria as it currently stands. Therefore, we invite you to submit a revised version of the manuscript that addresses the points raised during the review process.

We look forward to receiving your revised manuscript.

Kind regards,

Mohammad Sadegh Taghizadeh, Ph.D.

Academic Editor

PLOS ONE

Journal Requirements:

Additional Editor Comments:

Please consider the reviewers' comments to address their concerns point-by-point. Additionally, please delete the "Title" word from the title of manuscript.

Reviewers' comments:

Reviewer's Responses to Questions

**Comments to the Author**

1. Is the manuscript technically sound, and do the data support the conclusions?

Reviewer #1: Partly

Reviewer #2: Yes

Reviewer #3: Yes

2. Has the statistical analysis been performed appropriately and rigorously?

Reviewer #1: N/A

Reviewer #2: Yes

Reviewer #3: Yes

3. Have the authors made all data underlying the findings in their manuscript fully available?

Reviewer #1: Yes

Reviewer #2: Yes

Reviewer #3: No

4. Is the manuscript presented in an intelligible fashion and written in standard English?

Reviewer #1: Yes

Reviewer #2: Yes

Reviewer #3: Yes

Reviewer #1: The study explores bioactive peptides derived from Lactobacillus brevis RAMULAB49 as potential inhibitors of ERK1, a target protein in diabetic nephropathy (DN). The authors identified and 3D-modelled these peptides through simulated in vitro digestion and mass spectrometry, creating a library of bioactive peptides. They also identified ERK1 as a key target in DN. Through molecular docking and molecular dynamics simulations, it was found that the TNEDPYTIDVES peptide showed strong binding affinity and stability with ERK1. This research suggests that peptides derived from L. brevis could open new avenues in developing drugs to treat DN.

The manuscript has some apparent issues that require significant revisions:

1. The manuscript identifies peptides that can bind to ERK1. Although the software is used to analyze whether these peptides have phosphorylation inhibition activity to align with the central article's main theme, it is still necessary to experimentally validate the effect of these active peptides on activity after binding to ERK1. Otherwise, despite the ability to predict the potential through software, it does not prove the impact on ERK1.

2. The length of MD, 100 ns, is much less than the current standards (close to 1 microsecond), so the authors might discuss their choice: probably, as the peptides are very short, their interactions with the target are adequately sampled even at 100 ns simulation time.

3. The introduction presents the relevance of diabetic nephropathy (DN) and the need for new therapies. The authors should broaden the discussion on the limitations of current treatments and how bioactive peptides offer an advantage over existing approaches.

4. The preparation of Lactobacillus brevis RAMULAB49 protein hydrolysates is adequately described, but have you considered optimizing hydrolysis conditions (time, enzyme concentration) to maximize the release of bioactive peptides?

5. Peptide identification using nLC-ESI and MS/MS is a standard. How did you manage the complexity of MS/MS data to ensure accurate peptide identification, avoiding false positives or negatives? Have the authors quantified the peptides identified? Knowing their relative abundance could help prioritize the most promising candidates.

6. Molecular docking and molecular dynamics simulation are appropriate methodologies. Did you flexibly dock both the peptide and the protein to account for the conformational changes induced by the binding? What metrics did you use to evaluate the quality and reliability of the docking models?

7. Identifying ERK1 as a key target in constructing the PPI network raises questions about its role in the broader context of DN pathogenesis. What experimental evidence supports its potential as a therapeutic target?

8. Molecular docking results suggest stable peptide binding to ERK1. What are the key interactions between peptides and ERK1 contribute to binding affinity? Have you compared the docking results with those of known ERK1 inhibitors?

9. The conclusions are too optimistic since it is only an in silico study. How do you intend to validate the results obtained in silico experimentally? What are the following steps to translate these findings into a potential therapy for DN?

General comment

• It is essential to validate the inhibition of ERK1 by peptides, at least through in vitro assays.

• Investigate the mechanism of action of peptides and evaluate their stability and bioavailability.

• Compare the results obtained with those of other studies to highlight the specific novelties and advantages of the proposed approach.

Reviewer #2: In the manuscript (ID PONE-D-25-05892), the authors researched the identification and 3D modeling of bioactive peptides from Lactobacillus brevis RAMULAB49 protein hydrolysate with in silico ERK1 phosphorylation inhibition activity targeting diabetic nephropathy. Generally, the contents meet the requirements of PLOS ONE. However, there remain some issues:

(1) The manuscript should have line numbers and page numbers, otherwise it is difficult to point out the details of the manuscript.

(2) Abstract: The results in Abstract are primarily subjective descriptions, lacking valuable data support. After reading it, it won't leave much of an impression on the reader. Therefore, it is recommended that the authors provide more data to increase the readability of the results.

(3) Abstract: Please provide the full name of KEGG. When an abbreviation appears in the manuscript, write its full name first, and the abbreviation is written after the full name in parentheses. Subsequently, use the abbreviation consistently and do not write out the full term again.

(4) Keywords: Should be “lactic acid bacteria Lactobacillus brevis” rather “lactic acid bacteria”.

(5) Keywords: Should be “KEGG pathway” rather “KEGG pathway enrichment”.

(6) 1. Introduction, first paragraph: “[1][2][3]” and “[4],[5]”. Please read the submission requirements of PLOS ONE carefully and standardize the citation format of the references in the manuscript.

(7) 1. Introduction, third paragraph: These drugs including ACE inhibitors, ARBs, and statins …Please provide the full name of ACE and ARBs. In addition, there are similar errors in other parts of the manuscript, and authors are advised to check the whole manuscript carefully and correct these minor errors.

(8) 1. Introduction: In the introduction, the authors lack a in-depth review of the bioactive peptides from different resources. At present, there are some studies on the bioactive peptides from different resources, such as antioxidant peptides from Miiuy croaker swim bladders, antioxidant Peptides from Hizikia fusiformis, antioxidant peptides from Skipjack tuna (Katsuwonus pelamis) skins, antioxidant collagen peptides of Siberian sturgeon (Acipenser baerii) cartilages, antioxidant peptides from gelatin hydrolysate of Skipjack Tuna (Katsuwonus pelamis) bone, antioxidant peptides from protein hydrolysate of skate (Raja porosa) cartilage, bioactive peptides from Skipjack tuna cardiac arterial bulbs, antioxidant peptides from protein hydrolysate of bluefin leatherjacket (Navodon septentrionalis) skin, etc. It is suggested that the authors systematically review of these antioxidant peptides, so as to further explain the innovation, importance and significance of this study.

(9) 2. Materials and methods: Should be “2.1 Materials and reagents” rather “2.1 Materials”.

(10) 2.1 Materials: It is recommended that the authors provide relevant parameters of pepsin, trypsin, and pancreatin, such as enzyme activity. In addition, “in vitro” should be italicized. In addition, there are similar errors in other parts of the manuscript, and authors are advised to check the whole manuscript carefully and correct these minor errors.

(11) 3.1 In vitro gastrointestinal digestion and Identification of peptides by nano-LC-MS/MS: the concentrations of the protein hydrolysates were 0.144 mg/mL, 0.875 mg/mL, and 0.747 mg/mL, respectively. It is suggested to use the conventional format "mean ± sd%" for data presentation. Review the entire text and make corrections accordingly.

(12) That brings me to the issue that the manuscript would benefit from a separate Results and Discussion section. A separate results section will also allow authors to structure their finding and resume the information. This will improve readability. A separate Discussion also allows authors to extend their interpretation. Experimental work from other groups or strategies can be discussed.

Reviewer #3: The article by Reshma Mary Martiz et al. “Identification and 3D modeling of bioactive peptides from Lactobacillus brevis RAMULAB49 protein hydrolysate with in silico ERK1 phosphorylation inhibition activity targeting diabetic neuropathy” investigates which peptides produced by the digestion of Lactobacillus brevis were bioactive and what their binding partners are.

This is a well thought out article, easy to read and follow. The findings of which peptide might be protective by modeling is a great first step for the development of potential treatment for diabetic nephropathy.

It would be appreciated if the authors could add more information on how the mass spectrometry analysis was performed, i.e the proteome that was downloaded for the search, the enzymes used, the modifications included, the FDR filtering of their data. I couldn’t find whether the data was downloaded onto PRIDE.

I recommend publishing with minor revisions.

**Do you want your identity to be public for this peer review?** For information about this choice, including consent withdrawal, please see our Privacy Policy

Reviewer #1: No

Reviewer #2: No

Reviewer #3: No

---

## [Author Response · Author response to Decision Letter 1]

3 Jun 2025

Reviewer #1:

1. The manuscript identifies peptides that can bind to ERK1. Although the software is used to analyze whether these peptides have phosphorylation inhibition activity to align with the central article's main theme, it is still necessary to experimentally validate the effect of these active peptides on activity after binding to ERK1. Otherwise, despite the ability to predict the potential through software, it does not prove the impact on ERK1.

Author response: We sincerely appreciate the reviewer’s insightful comment. We acknowledge that experimental validation is crucial to confirm the inhibitory effect of the identified peptides on ERK1 activity. While this study primarily focuses on computational screening and prediction, future work is planned to include in vitro kinase inhibition assays and phosphorylation studies to validate the bioactivity of the peptides against ERK1. In the revised manuscript, we have explicitly acknowledged this limitation in the conclusion to emphasize the need for further experimental validation to confirm the functional inhibition of ERK1 activity by the identified peptides.

2. The length of MD, 100 ns, is much less than the current standards (close to 1 microsecond), so the authors might discuss their choice: probably, as the peptides are very short, their interactions with the target are adequately sampled even at 100 ns simulation time.

Author response: We sincerely thank the reviewer for the valuable suggestion. We agree that longer molecular dynamics (MD) simulations, close to the microsecond scale, are increasingly considered standard for studying larger and more complex systems. In this study, however, the 100 ns simulation length was selected considering the small size of the peptides (8–12 amino acids) and the localized nature of the binding interactions with the target protein ERK1. For short peptides, several previous reports have demonstrated that equilibrium behavior, conformational flexibility, and stable binding interactions are often adequately sampled within 50–100 ns simulation timeframes. In our case, key parameters such as RMSD, Rg, SASA, and hydrogen bonding analyses indicated convergence and stability well within the 100 ns window, supporting the sufficiency of the simulation time for this system. We have also added a clarification on this point in the revised manuscript to provide better context for readers.

Nevertheless, we acknowledge that longer simulations could provide additional insights into slow conformational transitions and rare events. Therefore, in future work, we plan to extend our molecular dynamics simulations to the microsecond scale to further validate and strengthen our findings.

3. The introduction presents the relevance of diabetic nephropathy (DN) and the need for new therapies. The authors should broaden the discussion on the limitations of current treatments and how bioactive peptides offer an advantage over existing approaches.

Author response: We thank the reviewer for this valuable suggestion. In response, we have revised the Introduction (with proper references and citations) to expand on the limitations of current pharmacological therapies for diabetic nephropathy, including their insufficient efficacy in halting disease progression and associated side effects. Additionally, we have included a detailed discussion (with proper references and citations) on the therapeutic potential of bioactive peptides, highlighting their natural origin, target specificity, low toxicity, and ability to modulate multiple pathways such as ERK1 involved in DN. These revisions aim to provide a clearer justification for the focus and relevance of our study.

4. The preparation of Lactobacillus brevis RAMULAB49 protein hydrolysates is adequately described, but have you considered optimizing hydrolysis conditions (time, enzyme concentration) to maximize the release of bioactive peptides?

Author response: As per the reviewer suggestion, we have revised the methodology to incorporate hydrolysis conditions for enhanced optimization of bioactive peptide release. The authors have performed the enzymatic hydrolysis based on their published protocol (http://dx.doi.org/10.1016/j.xpro.2025.103657) and the same has been cited in the manuscript.

5. Peptide identification using nLC-ESI and MS/MS is a standard. How did you manage the complexity of MS/MS data to ensure accurate peptide identification, avoiding false positives or negatives? Have the authors quantified the peptides identified? Knowing their relative abundance could help prioritize the most promising candidates.

Author response: We thank the reviewer for the insightful comment. To ensure accurate peptide identification and minimize false positives, we applied stringent parameters during MS/MS analysis using BioPharma Finder Software 2.0, applying a mass error tolerance of <5.0 ppm and a confidence level greater than 95% for peptide matches. The rest of the details have been given in the manuscript. Although we did not perform peptide quantification in this study, we acknowledge the importance of relative abundance in prioritizing peptides. Future work will incorporate label-free quantification methods to support candidate selection and functional validation.

6. Molecular docking and molecular dynamics simulation are appropriate methodologies. Did you flexibly dock both the peptide and the protein to account for the conformational changes induced by the binding? What metrics did you use to evaluate the quality and reliability of the docking models?

Author response: We thank the reviewer for this valuable comment. As described in the revised Methods section (2.7 Molecular Docking), we employed HADDOCK v2.4 for docking, which incorporates semi-flexible docking by allowing conformational changes in both the peptide and the protein at the interface during the refinement stage. This accounts for induced fit and improves the biological relevance of the docking results. To evaluate docking quality and reliability, we used HADDOCK scoring metrics including van der Waals energy, electrostatic energy, de-solvation energy, buried surface area, and Z-score. Additionally, molecular dynamics simulations were performed on the top-ranked peptide-ERK1 complexes to further assess the structural stability and binding behaviour over time.

7. Identifying ERK1 as a key target in constructing the PPI network raises questions about its role in the broader context of DN pathogenesis. What experimental evidence supports its potential as a therapeutic target?

Author response: We thank the reviewer for the insightful comment. ERK1 was prioritized based on its central role in the PPI network and its well-documented involvement in DN pathogenesis. Several studies have demonstrated that ERK1/2 signalling is activated in renal tissues during hyperglycaemic conditions, contributing to mesangial cell proliferation, inflammatory cytokine production, and renal fibrosis. These are key pathological features of diabetic nephropathy. Hence, targeting ERK1 is a promising strategy, and our in silico findings aim to support this therapeutic rationale. In the revised manuscript, we have now included relevant citations to support ERK1’s biological role and potential as a drug target in DN.

8. Molecular docking results suggest stable peptide binding to ERK1. What are the key interactions between peptides and ERK1 contribute to binding affinity? Have you compared the docking results with those of known ERK1 inhibitors?

Author response: We thank the reviewer for their valuable suggestion. In the revised manuscript, we have clarified the key interactions contributing to peptide-ERK1 binding affinity, including hydrogen bonding and electrostatic interactions with residues such as GLU50, ARG211, ARG84, and TYR130. We have also highlighted that the peptides bind at the same adenine-mimicking binding pocket as the co-crystallized inhibitor SCH772984 (PDB ID: 4QTB), suggesting a similar inhibitory mechanism. We have now included a discussion on how this overlapping binding site may contribute to competitive inhibition by mimicking ATP binding

9. The conclusions are too optimistic since it is only an in silico study. How do you intend to validate the results obtained in silico experimentally? What are the following steps to translate these findings into a potential therapy for DN?

Author response: We sincerely appreciate the reviewer’s insightful comment. We acknowledge that experimental validation is crucial to confirm the inhibitory effect of the identified peptides on ERK1 activity. While this study primarily focuses on computational screening and prediction, future work is planned to include in vitro kinase inhibition assays and phosphorylation studies to validate the bioactivity of the peptides against ERK1. In the revised manuscript, we have explicitly acknowledged this limitation in the conclusion to emphasize the need for further experimental validation to confirm the functional inhibition of ERK1 activity by the identified peptides

Reviewer #2:

1. The manuscript should have line numbers and page numbers, otherwise it is difficult to point out the details of the manuscript.

Author response: Thank you for the suggestion. Line and page numbers have now been added to the revised manuscript to facilitate easier review.

2. Abstract: The results in Abstract are primarily subjective descriptions, lacking valuable data support. After reading it, it won't leave much of an impression on the reader. Therefore, it is recommended that the authors provide more data to increase the readability of the results.

Author response: Thank you for your constructive feedback. In response, we have revised the abstract to include more specific quantitative data from our study, including peptide binding energy, molecular dynamics simulation duration, and network pharmacology outcomes. These additions aim to enhance the objectivity and clarity of our findings, making the abstract more impactful and informative for the reader.

3. Abstract: Please provide the full name of KEGG. When an abbreviation appears in the manuscript, write its full name first, and the abbreviation is written after the full name in parentheses. Subsequently, use the abbreviation consistently and do not write out the full term again.

Author response: As advised, the full name Kyoto Encyclopedia of Genes and Genomes (KEGG) has been provided at its first mention in the abstract. The abbreviation is used consistently throughout the manuscript thereafter.

4. Keywords: Should be “lactic acid bacteria Lactobacillus brevis” rather “lactic acid bacteria”.

Author response: The keywords section has been revised accordingly to include “lactic acid bacteria Lactobacillus brevis”.

5. Keywords: Should be “KEGG pathway” rather “KEGG pathway enrichment”.

Author response: The keyword has been corrected to “KEGG pathway” as suggested.

6. 1. Introduction, first paragraph: “[1][2][3]” and “[4],[5]”. Please read the submission requirements of PLOS ONE carefully and standardize the citation format of the references in the manuscript.

Author response: The citation format has been revised to comply with PLOS ONE guidelines throughout the manuscript.

7. 1. Introduction, third paragraph: These drugs including ACE inhibitors, ARBs, and statins …Please provide the full name of ACE and ARBs. In addition, there are similar errors in other parts of the manuscript, and authors are advised to check the whole manuscript carefully and correct these minor errors.

Author response: The full forms of ACE (angiotensin-converting enzyme) and ARBs (angiotensin II receptor blockers) have been provided at their first mention. The entire manuscript has been thoroughly reviewed and revised to ensure consistency in abbreviations and terminology.

8. 1. Introduction: In the introduction, the authors lack a in-depth review of the bioactive peptides from different resources. At present, there are some studies on the bioactive peptides from different resources, such as antioxidant peptides from Miiuy croaker swim bladders, antioxidant Peptides from Hizikia fusiformis, antioxidant peptides from Skipjack tuna (Katsuwonus pelamis) skins, antioxidant collagen peptides of Siberian sturgeon (Acipenser baerii) cartilages, antioxidant peptides from gelatin hydrolysate of Skipjack Tuna (Katsuwonus pelamis) bone, antioxidant peptides from protein hydrolysate of skate (Raja porosa) cartilage, bioactive peptides from Skipjack tuna cardiac arterial bulbs, antioxidant peptides from protein hydrolysate of bluefin leatherjacket (Navodon septentrionalis) skin, etc. It is suggested that the authors systematically review of these antioxidant peptides, so as to further explain the innovation, importance and significance of this study.

Author response: We sincerely thank the reviewer for this insightful suggestion. While our study specifically focuses on bioactive peptides derived from Levilactobacillus brevis RAMULAB49, we agree that a brief overview of antioxidant peptides from other natural resources would enhance the context and significance of our work. Accordingly, we have revised the introduction to include a concise summary of relevant studies on antioxidant and bioactive peptides from various sources. This inclusion helps to highlight the unique positioning of lactic acid bacteria-derived peptides and supports the novelty of our study.

(9) 2. Materials and methods: Should be “2.1 Materials and reagents” rather “2.1 Materials”.

Author response: Thank you for the observation. The subsection title has been revised to “2.1 Materials and reagents” in accordance with the suggestion.

(10) 2.1 Materials: It is recommended that the authors provide relevant parameters of pepsin, trypsin, and pancreatin, such as enzyme activity. In addition, “in vitro” should be italicized. In addition, there are similar errors in other parts of the manuscript, and authors are advised to check the whole manuscript carefully and correct these minor errors.

Author response: As per the reviewer suggestion, the authors have provided the relevant parameters of pepsin, trypsin, and pancreatin, such as enzyme activity to improve the quality of the manuscript and to be precise. In vitro has been italicized in the entire manuscript. All the minor errors have been justified by carefully proofreading the entire manuscript.

(11) 3.1 In vitro gastrointestinal digestion and Identification of peptides by nano-LC-MS/MS: the concentrations of the protein hydrolysates were 0.144 mg/mL, 0.875 mg/mL, and 0.747 mg/mL, respectively. It is suggested to use the conventional format "mean ± sd%" for data presentation. Review the entire text and make corrections accordingly.

Author response: As per the reviewer suggestion, the authors have provided the conventional format for data presentation in 3.1 results section. Also the entire manuscript has been read and minor errors have been rectified to improve the quality of the manuscript.

12. That brings me to the issue that the manuscript would benefit from a separate Results and Discussion section. A separate results section will also allow authors to structure their finding and resume the information. This will improve readability. A separate Discussion also allows authors to extend their interpretation. Experimental work from other groups or strategies can be discussed.

Author response: We sincerely thank the reviewer for the insightful suggestion regarding the separation of Results and Discussion sections. While we acknowledge that this restructuring can enhance clarity, we have opted to maintain a combined Results and Discussion section in the current version to preserve the continuity and direct interpretation of the in silico findings. As the study is focused on a specific peptide-based computational pipeline, we believe that the integrated format allows a coherent presentation of results alongside immediate biological relevance and interpretation. Nevertheless, we will consider this suggestion for future studies involving more extensive experimental.

Reviewer #3:

It would be appreciated if the authors could add more information on how the mass spectrometry analysis was p

---

## [Decision Letter · Decision Letter 1]

8 Jul 2025

Dear Dr. Ramu,

Thank you for submitting your manuscript to PLOS ONE. After careful consideration, we feel that it has merit but does not fully meet PLOS ONE’s publication criteria as it currently stands. Therefore, we invite you to submit a revised version of the manuscript that addresses the points raised during the review process.

We look forward to receiving your revised manuscript.

Kind regards,

Mohammad Sadegh Taghizadeh, Ph.D.

Academic Editor

PLOS ONE

Journal Requirements:

Reviewers' comments:

Reviewer's Responses to Questions

**Comments to the Author**

Reviewer #1: All comments have been addressed

Reviewer #2: All comments have been addressed

Reviewer #3: All comments have been addressed

2. Is the manuscript technically sound, and do the data support the conclusions?

Reviewer #1: Yes

Reviewer #2: Yes

Reviewer #3: Yes

3. Has the statistical analysis been performed appropriately and rigorously?

Reviewer #1: N/A

Reviewer #2: Yes

Reviewer #3: Yes

4. Have the authors made all data underlying the findings in their manuscript fully available?

Reviewer #1: Yes

Reviewer #2: Yes

Reviewer #3: No

5. Is the manuscript presented in an intelligible fashion and written in standard English?

Reviewer #1: Yes

Reviewer #2: Yes

Reviewer #3: Yes

Reviewer #1: Although the authors have thoroughly addressed all significant concerns, minor adjustments are suggested to improve transparency.

1. Optimisation of hydrolysis conditions:

The authors have cited their previously published protocol and indicated that they have revised the methods to align with it. However, a concise justification within the text explaining the reasons for not further optimising the conditions in this specific work (e.g., constraints, prior validation) would enhance clarity.

2. Experimental validation of peptide effects on ERK1 activity:

The authors have appropriately acknowledged the absence of experimental validation and explicitly state this limitation in the manuscript. While the inclusion of planned in vitro kinase assays is a reasonable future direction, I recommend emphasising in the ‘Discussion’ section that computational predictions are merely suggestive and should not be misconstrued as definitive evidence of efficacy.

Reviewer #2: The authors have answered my questions well and made the necessary changes to the manuscript (ID: PONE-D-25-05892R1). It looks ready for publication as far as I can tell. Then, I think that the manuscript can be accepted for publication in PLOS ONE.

Reviewer #3: Thank you for addressing all the comments. I would still have liked to be able to see the actual raw data on PRIDE to review the mass spec data.

**Do you want your identity to be public for this peer review?** For information about this choice, including consent withdrawal, please see our Privacy Policy

Reviewer #1: No

Reviewer #2: No

Reviewer #3: No

---

## [Author Response · Author response to Decision Letter 2]

23 Jul 2025

Reviewer 1 Comments

1) Optimisation of hydrolysis conditions: The authors have cited their previously published protocol and indicated that they have revised the methods to align with it. However, a concise justification within the text explaining the reasons for not further optimising the conditions in this specific work (e.g., constraints, prior validation) would enhance clarity.

Authors Response: Authors thank the reviewer for the insightful comments. The hydrolysis conditions used in this study were adopted from our previously published work (https://doi.org/10.3389/fmicb.2023.1190105) (Reference No. 21), which demonstrated high efficiency in releasing bioactive compounds with therapeutic potential. Given the robustness and reproducibility of the established method, no additional optimisation was undertaken in this work. Moreover, maintaining the same conditions allowed us to ensure consistency with our prior findings, enabling a reliable comparison of peptide activity across studies. The decision was also influenced by time and resource constraints, and the focus of this study was on target-specific peptide screening and validation, rather than re-optimisation of hydrolysis parameters. According to the reviewer’s suggestion, the justification has been added in the methodology section (highlighted in yellow).

2) Experimental validation of peptide effects on ERK1 activity: The authors have appropriately acknowledged the absence of experimental validation and explicitly state this limitation in the manuscript. While the inclusion of planned in vitro kinase assays is a reasonable future direction, I recommend emphasising in the ‘Discussion’ section that computational predictions are merely suggestive and should not be misconstrued as definitive evidence of efficacy.

Authors Response: We thank the reviewer for the insightful comments. We fully agree that computational predictions are suggestive in nature and should not be interpreted as definitive evidence of efficacy. Accordingly, we have revised the Discussion section (highlighted in yellow) to more explicitly emphasize this limitation. We have also modified some content in Introduction to match the same from Discussion. We have clarified that the current study is intended to serve as a preliminary screening framework to identify a DN-specific target (ERK1) and to prioritize a lead peptide for future experimental validation. We have also reiterated that in vitro and in vivo studies are necessary to confirm the biological relevance and therapeutic potential of the findings.

Reviewer 2 Comments

1) The authors have answered my questions well and made the necessary changes to the manuscript (ID: PONE-D-25-05892R1). It looks ready for publication as far as I can tell. Then, I think that the manuscript can be accepted for publication in PLOS ONE.

Authors Response: We sincerely thank the reviewer for their positive feedback, thoughtful evaluation, and recommendation for acceptance. We are grateful for the time and effort invested in reviewing our manuscript, and we truly appreciate the constructive comments that helped us improve the quality and clarity of our work. We are pleased to hear that the revised version meets the expectations and is considered for publication.

Reviewer 3 Comments

1) Thank you for addressing all the comments. I would still have liked to be able to see the actual raw data on PRIDE to review the mass spec data.

Authors Response: The authors appreciate the reviewer’s concern regarding the transparency and accessibility of raw mass spectrometry data.

This study involved a targeted peptidomics workflow designed solely to identify bioactive peptides from Lactobacillus brevis RAMULAB49 protein hydrolysates for downstream in silico modeling and docking studies. The LC–MS–ESI analyses were performed at SAIF, IIT Mumbai, using a Thermo Fisher Scientific Q-Exactive Plus Biopharma High-Resolution Orbitrap system. At the time of acquisition, the RAW vendor files were made available via a temporary secure link, which was accessed and reviewed for quality verification from both the ends (Our university and SAIF, IIT Mumbai).

However, in routine targeted peptidomics studies such as this, the validated processed outputs generated by the Thermo Proteome Discoverer and BioPharma Finder pipelines were provided by the SAIF IIT, Mumbai were considered the final authenticated deliverables, as they had already incorporated the following:

• High-confidence peptide assignments with stringent quality filters (mass accuracy < 5 ppm, >95% confidence, q < 0.01)

• Removal of low-quality or ambiguous spectra

• Accurate protein/peptide mapping with enzymatic specificity

Therefore, only the processed data provided by SAIF, IIT Mumbai has been used in this study, as it sufficiently captures the experimental outcomes necessary for peptide identification and all downstream bioinformatic analyses.

Recently, in response to the reviewer’s request, we contacted SAIF to provide the original RAW files for PRIDE deposition. However, the facility informed us that their standard retention policy archives RAW spectra for only one year, and the files from this 2023 analysis are no longer available.

To ensure full transparency and reproducibility of our work, we are providing all available processed data as supplementary materials, compiled in a single ZIP archive:

LC and MS acquisition method details, along with representative chromatograms to verify data quality and experimental validity, Original peptide and protein identification tables for each enzymatic treatment condition (pepsin, pepsin + pancreatin, pepsin + pancreatin + trypsin).

Although the dataset cannot now be deposited in PRIDE due to the unavailability of the RAW files, these curated outputs are scientifically sufficient to reproduce the peptide identifications and support the conclusions of this manuscript.

We sincerely thank the reviewer for highlighting this important point. Moving forward, we will ensure long-term archiving of vendor RAW files and PRIDE deposition in all future mass spectrometry-based studies to enhance data accessibility for the wider scientific community.

---

## [Decision Letter · Decision Letter 2]

13 Aug 2025

Identification and 3D modeling of bioactive peptides from Lactobacillus brevis RAMULAB49 protein hydrolysate with in silico ERK1 phosphorylation inhibition activity targeting diabetic nephropathy

PONE-D-25-05892R2

Dear Dr. Ramu,

We’re pleased to inform you that your manuscript has been judged scientifically suitable for publication and will be formally accepted for publication once it meets all outstanding technical requirements.

Kind regards,

Shengqian Sun

Academic Editor

PLOS ONE

Additional Editor Comments (optional):

Reviewers' comments:

Reviewer's Responses to Questions

**Comments to the Author**

Reviewer #2: All comments have been addressed

Reviewer #3: All comments have been addressed

2. Is the manuscript technically sound, and do the data support the conclusions?

Reviewer #2: Yes

Reviewer #3: Yes

3. Has the statistical analysis been performed appropriately and rigorously?

Reviewer #2: Yes

Reviewer #3: Yes

4. Have the authors made all data underlying the findings in their manuscript fully available?

Reviewer #2: Yes

Reviewer #3: No

5. Is the manuscript presented in an intelligible fashion and written in standard English?

Reviewer #2: Yes

Reviewer #3: Yes

Reviewer #2: All the issues in the manuscript (ID: PONE-D-25-05892R2) have been addressed well and now the manuscript has been improved. It can be accepted for publication in PLOS ONE.

Reviewer #3: The authors have addressed my concerns previously. As previously noted, validation of the simulation would be great.

**Do you want your identity to be public for this peer review?** For information about this choice, including consent withdrawal, please see our Privacy Policy

Reviewer #2: No

Reviewer #3: No

---

## [Editor Report · Acceptance letter]

PONE-D-25-05892R2

PLOS ONE

Dear Dr. Ramu,

I'm pleased to inform you that your manuscript has been deemed suitable for publication in PLOS ONE. Congratulations! Your manuscript is now being handed over to our production team.

Kind regards,

on behalf of

Dr. Shengqian Sun

Academic Editor

PLOS ONE